# Thermodynamic framework for assessing dissolutive wetting behaviors in metallic systems

Youqing Sun [1,2,3], Shoufeng Yang [1,2] ✉, Xinghong Cai [3,4] ✉, Zhongfu Cheng [3] ✉, Wantong Chen [3], Nele Moelans[3], Muxing Guo[3] & David Seveno [3]

Despite its significance in practical applications, until today dissolutive wetting in metallic systems remains poorly understood due to the intricate liquid/solid interactions. In this study, we investigate nine metallic systems encompassing a vast range of thermodynamic behaviors. Our findings reveal three distinct wetting behaviors and identify two key thermodynamic parameters that control solid solution formation and then dissolutive wetting behaviors, namely the solubility of the substrate element in the liquid ($C_L$) and the solidification composition range ($C_R$). We observe a slow spreading due to the step flow mechanism in systems with a relatively low $C_L$ and narrow $C_R$. For systems with high $C_L$, the spreading begins relatively fast and transitions to the step flow mechanism when $C_L$ is reached. Systems with a large $C_R$, conversely, exhibit consistently fast spreading. Density functional theory (DFT) calculations provide deeper insights into the underlying atomistic mechanisms affecting the solid solution formation. Our results offer inspiration for optimizing high-temperature processing techniques, such as welding, coating and high temperature infiltrating.

Dissolutive wetting of liquid metals on solid metals is crucial for multiple high-temperature processes, including various welding methods[1,2], multiple surface treatment technologies[3], steelmaking[4,5], liquid phase sintering[6], metal recycling[7], and high-temperature infiltrating[8,9], etc. Wetting properties then directly determine the final quality of products produced from these processes, such as automotive components[10], electrical appliances[11,12], carbide blades[13], recycled metals[14], and metal matrix composites[15], etc. However, dissolutive wetting at high temperatures is not fully understood due to complicated physicochemical activities arising from liquid (L)/solid (S) interfaces, including diffusion[16], dissolution[17,18] and so on. Compared with product forming systems with intermetallic compounds (IMCs) forming at the L/S interface, dissolution is the primary interfacial

activity in dissolutive systems[19]. Dissolutive wetting behavior can be further categorized into two main types[20]: (1) purely dissolutive systems, where dissolution is the sole liquid-solid interaction (e.g., Sn/Bi[17] and Cu/Ni[21]); (2) product forming systems at temperatures exceeding the stability limits of all products (e.g., Cu/Ti at 1060 °C[22] and Cu/Si at 1100 °C[23]).

The product forming systems have been extensively analyzed in the literature and well described by reaction-limited[24–26], diffusion-limited[27,28], and ridge-controlled[29] models. However, the literature about dissolutive wetting is limited and raises contradictory conclusions. In some studies, the dissolution at the L/S interface exerts minimal effects on wetting[22,30–32], instead, the liquid/solid friction at the contact line (CL) and hydrodynamic viscous dissipation in the bulk

[1]Chongqing Institute of Green and Intelligent Technology, Chinese Academy of Sciences, Chongqing 400714, China. [2]University of Chinese Academy of Sciences, Chongqing School, Chongqing 400714, PR China. [3]Department of Materials Engineering, KU Leuven, Kasteelpark Arenberg 44, 3001 Leuven, Belgium. [4]Chongqing Key Laboratory for Advanced Materials and Technologies of Clean Energies, School of Materials and Energy, Southwest University, Chongqing 400715, China. ✉e-mail: shoufeng.yang@cigit.ac.cn; caixinghong@cqust.edu.cn; zhongfu.cheng@kuleuven.be

are the dominant factors[22,32]. In some other studies, dissolution is found to affect the L/S interfacial geometries[23,33], alter the interface/surface energy[34], induce a Marangoni flow[35,36] or trigger the formation of a precursor film ahead of the CL[37]. In our previous study focusing on dissolutive Cu/Ni systems[21], we proposed that the CL advancement is controlled by a new step flow mechanism for which all these factors coexist. Specifically, the CL is initially pinned by the solid solution layer formed at L/S interfaces and extend to the CL. Subsequently, a Ni-concentration induced Marangoni flow drives the liquid toward the CL. The Marangoni convection, in conjunction with enhanced affinity between the liquid Cu and Ni substrates, leads to the formation of a thin precursor film ahead of the CL. As liquid continuously accumulates at the CL, the droplet CL eventually overcomes the physical barrier and suddenly overflows. This process repeats, resulting in the progressive advancement of the CL.

The step flow mechanism reveals the significance of substrate dissolution and may also be applicable for other dissolutive systems. Additionally, we believe that the contradictory conclusion regarding the effects of substrate dissolution may arise from the different dissolution behaviors (e.g., solubility of substrate element in liquids) in various dissolutive systems. In this regard, we address the following questions: How do different dissolution behaviors of solid metals into liquid metals affect wetting behaviors? Can we assess dissolutive wetting behaviors in metallic systems based on the dissolution behavior? Our overarching objective is to unravel the mechanisms controlling wetting behaviors in dissolutive metallic systems.

Here, we observe the spreading behavior of nine different dissolutive metallic systems (Cu/Ti, Cu/Ni, Cu/Fe, Au/Pt, Ag/Ni and Ag/Ti at 1150 °C; Sn/Cu at 800 °C; Ag/Cu at 1000 °C; and Sn/Fe at 950 °C) using in-situ observation and non-contact heating sessile drop experiments. Dissolution and its effects on spreading are investigated by quenching L/S interfaces and thermodynamic modeling. Furthermore, Density functional theory (DFT) calculations are employed to unveil the mechanism for the solid solution formation at a fundamental level (e.g., charge density, density of states and bond strengths). We found that wetting behaviors in these nine systems strongly correlate with two thermodynamic parameters: the solubility of the substrate metal within the liquid metal ($C_L$) and the solidification composition range ($C_R$) between the liquidus and solidus line at a given temperature. Specifically: (1) wetting systems with limited $C_L$ and $C_R$ can rapidly induce solid solution formation at the L/S interface inducing a step flow behavior at early stage, resulting in an apparent slow spreading dynamic; (2) For systems with a limited $C_R$ but a high $C_L$, the spreading is firstly fast and then controlled by the step flow mechanism at the later stage after reaching the $C_L$; (3) Only a fast spreading dynamic is expected for systems with large $C_R$. In other words, a limited $C_R$ is required for the step flow mechanism and slow spreading, while $C_L$ controls its onset, with lower $C_L$ leading to a quicker initiation.

## Results
### Step flow behaviors
The CL movement in different nine systems is in-situ observed through a Confocal Scanning Laser Microscope (CSLM). These nine systems can be categorized into three cases according to the combination of $C_L$ and $C_R$. For instance, at 1150 °C, the Cu/Ni system exhibits a small Ni solubility with $C_L$ = 12.0 at. % and $C_R$ = 5.2 at. % (Case 1). In contrast, the Cu/Ti system has a large Ti solubility, $C_L$ = 71.8 at. % while maintaining a modest $C_R$ = 16.8 at. % (Case 2). Finally, the Cu/Fe system presents a minimal Fe solubility, $C_L$ = 4.7 at. % but a significant $C_R$ = 83.9 at. % (Case 3). A combination of large $C_L$ and $C_R$ is thermodynamically impossible, the systems investigated then encompass nearly all possible dissolution scenarios for metallic systems. Fig. 1a indicates the traveling distance of the CL during the isothermal stage (contact heating) for Cu/Ni, Cu/Ti, and Cu/Fe systems at 1150 °C, while results

for the remaining systems corresponding to Cases 1, 2, and 3 are provided in the supplementary information. It should be noted that Cu droplets rapidly complete spreading on Fe substrates upon melting (beyond the temporal resolution of CSLM), so no CL advancement was observed during the isothermal stage. As expected, these systems show different spreading behaviors largely determined by the occurrence of CL jumps resulting from the step flow mechanism, schematically illustrated by Fig. 1b. It involves (i) the pinning of the CL by the solidified edge due to the formation of a solid solution layer at L/S interfaces, (ii) a liquid flow towards the edge as a result of an induced Marangoni flow attributed to a surface tension gradient, (iii) the formation of a precursor film caused by enhanced affinity between liquids and solids, and finally (iv) a sudden overflow due to the dynamic accumulation of liquids at the edge. More details about step flow behaviors for the Cu/Ni system have been given in our previous study[21,38]. For instance, the formation of a solid solution layer was confirmed by in-situ observation and quenching[21]. The effects of dissolution on interfacial energy, Marangoni flow and interface morphology were characterized experimentally and further investigated by CFD simulations[38]. The step flow mechanisms are observed for Cu/Ni (Fig. 1c, Supplementary Fig. 1a, Supplementary Movie 1) and Cu/Ti (Fig. 1d, Supplementary Fig. 1b, Supplementary Movie 2) systems, but not for Cu/Fe system (as confirmed by the almost unchanged CL position, Fig. 1e, Supplementary Fig. 1c). It is interesting to note that the step flow mechanism initiates early after the beginning of the isothermal stage for the Cu/Ni system, while it emerges after ~2000 s for the Cu/Ti system (experiments are repeated three times as shown in Supplementary Fig. 2a). In conclusion, the spreading behaviors of these metallic systems can be categorized into three cases:

- Spreading in the step flow mechanism at the beginning of the isothermal stage for Cu/Ni (Case 1),
- Spreading in the step flow mechanism after a period of time (~2000 s) for Cu/Ti (Case 2),
- Continuous rapid spreading without any steps for Cu/Fe (Case 3).

### Quenched microstructures
Fig. 2 shows the top-view secondary electron (SE) pictures of quenched microstructures for Cu/Ni, Cu/Ti, and Cu/Fe systems, revealing flattened Cu droplets for all three substrates (Fig. 2a, Supplementary Fig. 3a), suggesting good wettability. The microstructure of repeated Cu/Ni experiments is detailed in Supplementary Fig. 2b–e. The enlarged SE pictures near the CL (Fig. 2b, Supplementary Fig. 3b) show that sharp edges exist in the Cu/Ni and Cu/Ti systems, but are not evident in the Cu/Fe system. The element point analysis near the CL indicate different distributions of dissolved substrate elements within the Cu droplets, specifically, a Ni composition of 15.9 at. % (spot 1 in Fig. 2b and e), a large Ti composition of 84.6 at. % (spot 2 in Fig. 2b and f) and a minor Fe composition of 5.4 at. % (spot 3 in Fig. 2b and g). This is further supported by the Energy dispersive spectroscopy (EDS) mapping analysis (Fig. 2c, d).

The formation of sharp edges near the CL is further validated both from a side or cross-section view (Fig. 3a). It can be seen that a sharp edge with a thickness of 21 μm forms at the CL in the Cu/Ni system (the side-view SE picture, Fig. 3b, Supplementary Fig. 3c). The EDS point analysis reveals that the edge is Ni-rich, with a composition of 55.3 at. % (spot 4 in Fig. 3b and g). The cross-section backscattered electron (BSE) picture (Fig. 3c) and its corresponding EDS mapping analysis (Fig. 3d) confirm that Ni distributes along the L/S interface and extends towards the CL. For Cu/Ti systems (the side-view SE picture, Fig. 3e), a sharp edge (11 μm) with a Ti composition of 90.2 at. % (spot 5 in Fig. 3e and h) is also found at the CL. In contrast, no conspicuous sharp edge is observed for Cu/Fe systems (the side-view SE picture, Fig. 3f). In conclusion, these metallic systems can also be divided into three cases

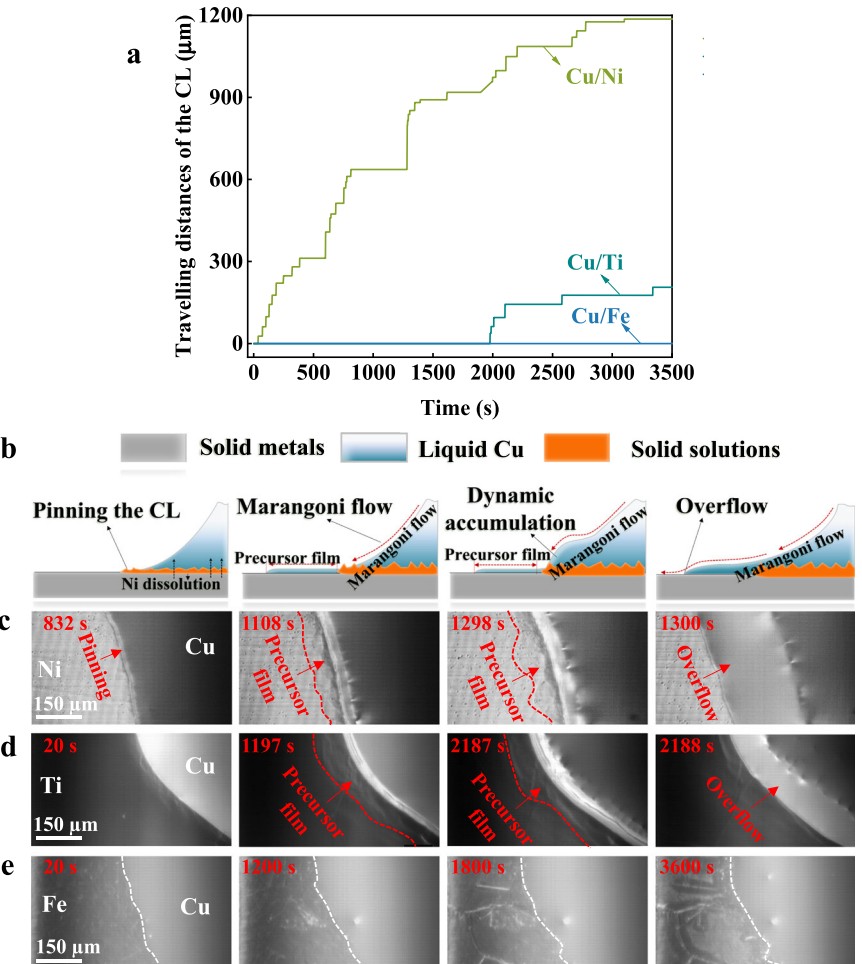

**Fig. 1 | Step flow behaviors in Cu/Ni, Cu/Ti and Cu/Fe systems at 1150 °C.**
**a** Traveling distances of the contact line (CL) during the isothermal stage (contact heating methods) for the three systems. **b** Illustration of the step flow mechanisms for (**c**) Cu/Ni and (**d**) Cu/Ti, while (**e**) the Cu/Fe system does not show evidence of a step flow. The blue, gray and orange colors in (**b**) indicate liquid metals, solid substrates and solid solutions, respectively. The red and white dash lines represent the endpoints of the precursor film and the CL, respectively. Source data for (**a**) are provided as a Source data file.

according to the existence of sharp edges near the CL and the distribution level of substrate elements near the edge:

- Cu/Ni shows obvious sharp edges with a certain amount of substrate element (Case 1),
- Cu/Ti exhibits the same characteristic of sharp edges but with substantial substrate element (Case 2),
- Cu/Fe is characterized by the absence of sharp edges and relatively insignificant presence of substrate element (Case 3).

### Thermodynamic perspective

It seems that these various metallic systems can be categorized as different cases based on the introduction of the step flow mechanism, the presence of sharp edges at the CL, and the distribution level of substrate elements. This also suggests that these three factors may be closely interconnected with each other. We believe that different amounts of the substrate metal within the liquid and the generation of sharp edges are related to diverse dissolution and solidification behaviors. Phase diagrams of Cu/Ni (Fig. 4a, b), Cu/Ti (Fig. 4c), and Cu/Fe (Fig. 4d) systems are then calculated with the red ($C_L$) and black lines representing the liquidus and solidus lines, respectively. $C_R$ represents the compositional range characterized by the coexistence of liquid and solid phases[39]. Assuming that equilibrium solidification occurs at the liquid metal/solid metal interface according to the Scheil

equation[40], the fractions of liquids and solid solutions in $C_R$ can be calculated using the level rule[39]. A small $C_R$ value indicates that the liquid can solidify rapidly and completely when the dissolved substrate element exceeds $C_L$. The Cu/Ni system (Case 1) with a low Ni solubility (12.0 at. %) and small $C_R$ (5.2 at. %) leads to a quick Ni saturation and the rapid formation of the solid solution layer at the L/S interface. This promptly formed layer serves as a solidified edge, pinning the droplet CL and then triggering a step flow behavior during the first moment of the isothermal stage. The quenched Ni composition at spot 4 (Fig. 3b) confirms that the sharp edge is formed due to an isothermal solidification (Fig. 4a, b). The fraction of solid solutions reaches 75% (spot 1 in Fig. 2b) even on the liquid surface near the CL. For the Cu/Ti system (Case 2), the Ti solubility increases to 71.8 at. %, while $C_R$ remains modest at 16.8 at. %. Consequently, saturation takes more time, but complete solidification can still occur after saturation due to low $C_R$ (Fig. 4c). The droplet partially solidifies (75% solid solutions) with a Ti composition of 84.6 at. % (spot 2 in Fig. 2b) at the L/G interface near the CL and completely solidifies at the position of the sharp edge with a higher Ti composition of 90.2 at. % (spot 5 in Fig. 3e). A step flow behavior is observed after 2000 s when the formation of solid solutions initiates. For the Cu/Fe system (Case 3), despite a low Fe solubility of 4.7 at. %, the high $C_R$ (89.2 at. %) impedes the complete solidification even after reaching saturation (Fig. 4d). At the CL, only 0.8% solids form with a Fe composition of 5.4 at. % (spot 3 in Fig. 2b), while complete solidification requires a threshold $C_R$ of 2 at. %

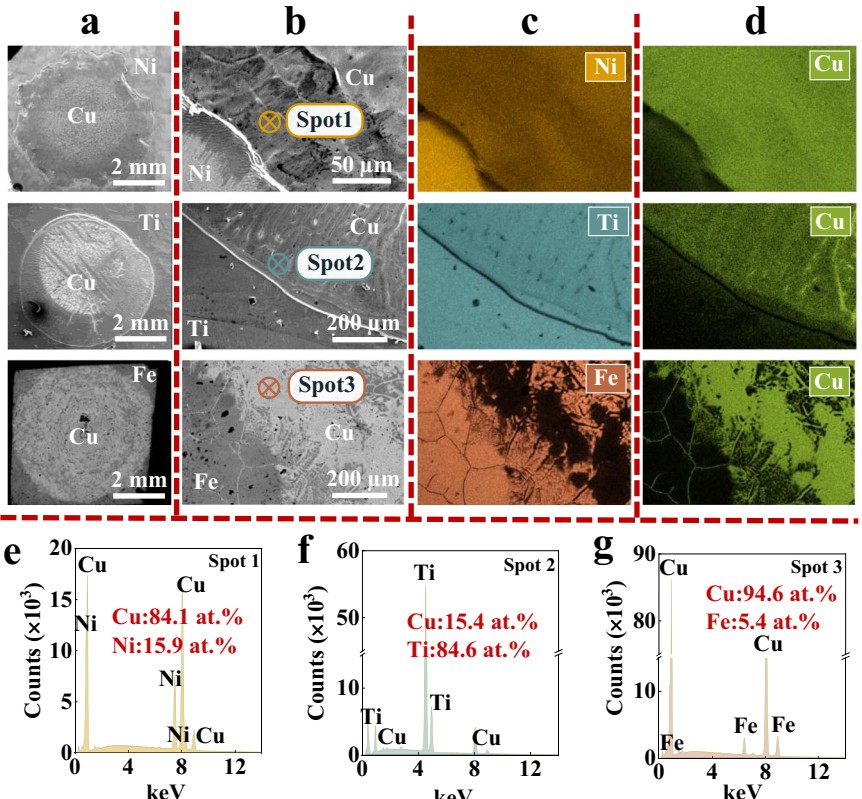

**Fig. 2 | Quenched microstructures of the Cu/Ni, Cu/Ti and Cu/Fe systems. a** Top-view secondary electron (SE) images of quenched Cu droplets on Ni, Ti and Fe substrates. **b** Enlarged SE images near the contact line (CL) indicating the presence of sharp edges for Cu/Ni and Cu/Ti, but not for Cu/Fe systems. The energy dispersive spectroscopy (EDS) mapping pictures reveal (**c**) a significant Ni, an even more substantial Ti, but a minimal Fe distribution within (**d**) the Cu droplets. **e–g** EDS point analysis acquired at the liquid/gas interface near the CL, corresponding to spots 1–3 marked in (**b**). Source data for (**e–g**) are provided as a Source data file.

(Supplementary Fig. 4). As a result, no step flow is observed in the Case 3. In conclusion, $C_R$ determines the occurrence of step flow mechanism, while $C_L$ controls its timing.

### Atomistic interactions

The formation of solid solutions, controlled at the fundamental level by atomistic interactions, is one of the primary factors controlling the step flow mechanism. To gain insights into the underlying atomistic mechanisms controlling the solid solution formation, we therefore implemented DFT numerical simulations[41] to complement our experimental data and thermodynamic calculations for Cu/Ni, Cu/Ti, and Cu/Fe systems. The similar atom size and lattice structure of Cu and Ni promote the formation of substitutional solid solution[42]. Cu/Ti and Cu/Fe systems are also reported to predominantly form substitutional solid solutions, though interstitial solid solutions may arise under special conditions, such as rapid cooling[43,44]. We constructed models by substituting Cu atoms with Ni, Fe and Ti solute atoms in an FCC Cu crystal to simulate the solid solution formation. The atomistic interactions and the consequent energy change which cannot be determined neither experimentally nor by thermodynamic calculations (see previous section) are our main focuses. It is reported heats of solution ($H_S$) for substituting solute atoms in Cu crystal can be split into a structural contribution ($H_{SC}$) caused by the lattice distortion and a chemical contribution ($H_{CC}$) resulting from solvent-solute interatomic interactions (Eq. 1)[45,46].

$$H_S = H_{SC} + H_{CC} \qquad (1)$$

Figure 5a–d schematically illustrates the methodology to predict $H_{SC}$ and $H_{CC}$. $H_{SC}$ is comprised of energy changes arising from the lattice distortion when removing a solvent Cu atom ($E_{HR}$) and substituting a solute atom (Ti, Ni, or Fe) with a different radii ($E_{SS}$) (Eq. 2)[46]. In practice, a Cu supercell is first relaxed, leading to an energy change $E_A$ (Fig. 5a). Creating an atom vacancy by removing a Cu atom, the energy $E_B$ of this system can be obtained (Fig. 5b). The vacancy is then filled by every type of substituted solute atom (Fig. 5c) forming bonds with surrounding Cu atoms (Fig. 5d). To determine the energy change due to the substitution, the solute atom is removed from the lattice structure and the energy change $E_C$ calculated. In other words, $E_C$ is the reversed energy change caused by replacing the vacancy with a solute atom. $E_{HR}$ and $E_{SS}$ can be then calculated using Eqs. 3 and 4, respectively.

$$H_{SC} = E_{HR} + E_{SS} \qquad (2)$$

$$E_{HR} = E_B - E_A + \sigma_{Cu} \qquad (3)$$

$$E_{SS} = E_C - E_B \qquad (4)$$

where $\sigma_{Cu}$ is the energy of per Cu atom in the Cu supercell.

The chemical contribution $H_{CC}$ is determined by the chemical effect caused by the Cu-Ti (Ni or Fe) interactions[47]. It can be calculated by Eq. 5.

$$H_{CC} = E_D - E_C - \sigma_{Ni/Ti/Fe} \qquad (5)$$

where $E_D$ is the energy of the relaxed Cu supercell containing one solute atom type. $\sigma_{Ti/Fe/Ni}$ is the energy of per solute atom in the solute element supercell.

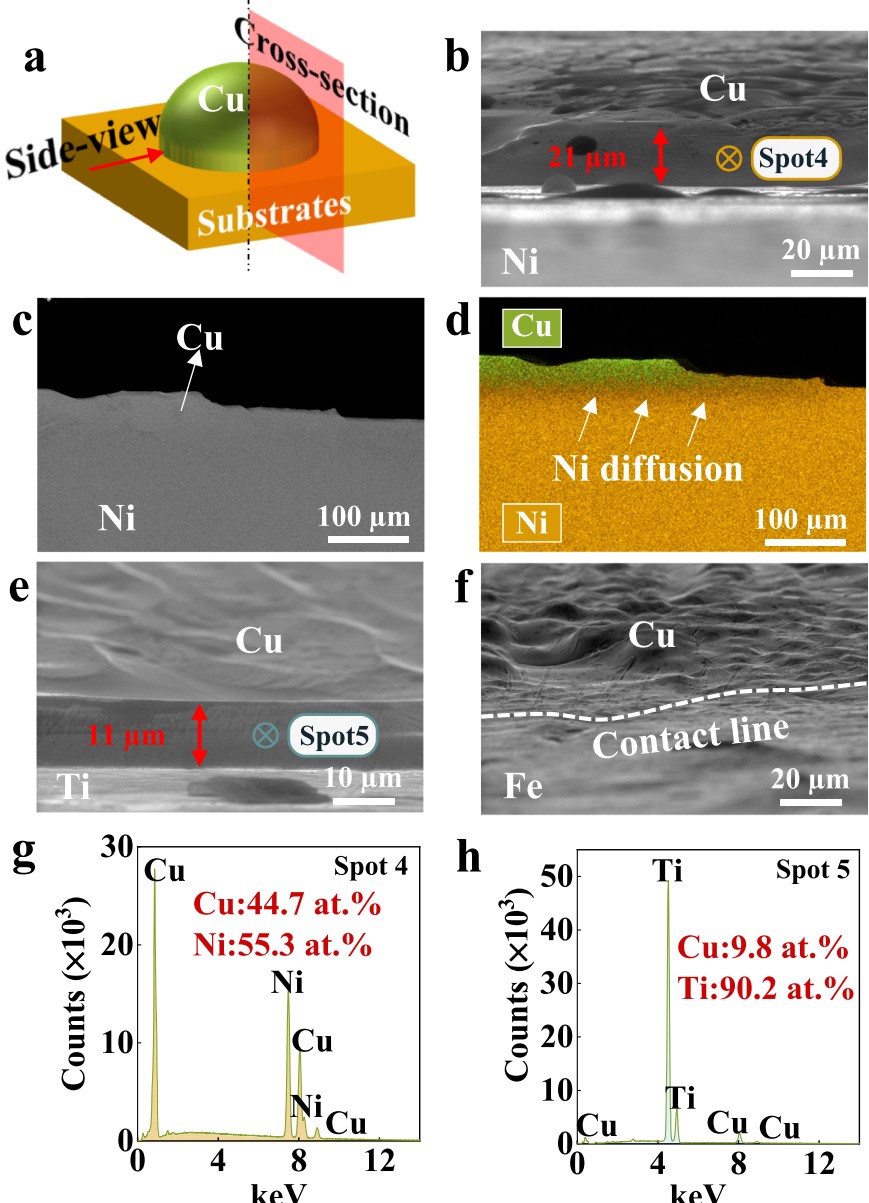

**Fig. 3 | side-view and cross-section microstructures of the Cu/Ni, Cu/Ti and Cu/Fe systems. a** Schematic of the side-view and cross-section geometries. **b** The side-view secondary electron (SE) image, (**c**) the cross-section backscattered electron (BSE) image and (**d**) the energy dispersive spectroscopy (EDS) mapping picture confirm the formation of a sharp edge in the Cu/Ni system. The sharp edges can also be observed from (**e**) the side-view SE picture for Cu/Ti, but not observed from (**f**) the side-view SE picture for Cu/Fe. The white dash line in (**f**) represents the contact line. **g**, **h** EDS point analysis at the sharp-edge regions, corresponding to spots 4 and 5 marked in (**b**) and (**e**), respectively. Source data for (**g**, **h**) are provided as a Source data file.

Table 1 indicates $H_S$ for different systems. To further decipher the chemical contribution, planar charge density distributions were calculated for Cu/Ti (Fig. 5e), Cu/Ni (Fig. 5f), and Cu/Fe (Fig. 5g). The local electron density near each solute atom is compared in Fig. 5h. Additionally, density of states (DOS) near the Fermi level (Fig. 5i–k) were extracted for all systems, showing overlapping Cu-3d and solute-3d orbitals in the energy range of −5.0 to −2.0 eV. The magnitude of the DOS peak in this energy range follows Ti > Ni > Fe.

DFT results indicate that Cu atoms are more easily substituted by Ti than by Ni or Fe, as confirmed by the order of $H_S$: Cu/Ti (0.03 eV) < Cu/Ni (0.28 eV) < Cu/Fe (0.65 eV). Consequently, the Cu crystal can accommodate more Ti atoms than Ni, and Fe atoms can hardly replace Cu to form solid solutions. This trend is confirmed by experimental observations, showing that the Ti concentration (spot 5 in Fig. 3e) is higher than the Ni concentration (spot 4 in Fig. 3b) in solid solutions, while only a limited Fe concentration (spot 3 in Fig. 2b) is observed in the solid solution formation in the Cu/Fe system. This large difference is primarily due to the chemical contribution ($H_{CC}$) but not the structural contribution ($H_{SC}$) according to Table 1. In other words, the bonding between the solute atom and neighboring host Cu atoms governs substitution energetics more than the lattice distortion. The higher charge density in the vicinity of the Ti atom than that of the Ni and Fe atom indicates a stronger Cu-Ti bond compared to the Cu-Ni and Cu-Fe bonds. The order of the magnitude of the DOS peak characterizes the chemical bond strength formed by orbital hybridization, i.e., Cu-Ti > Cu-Ni > Cu-Fe. The higher Ti concentration in Cu-Ti solid solutions is then due to stronger metallic bonds with Ti-3d orbitals significantly contributing to the DOS, leading to an improved conductivity and enhancing charge transport. We acknowledge that our substitutional solute model ignores factors such as multi solute atoms

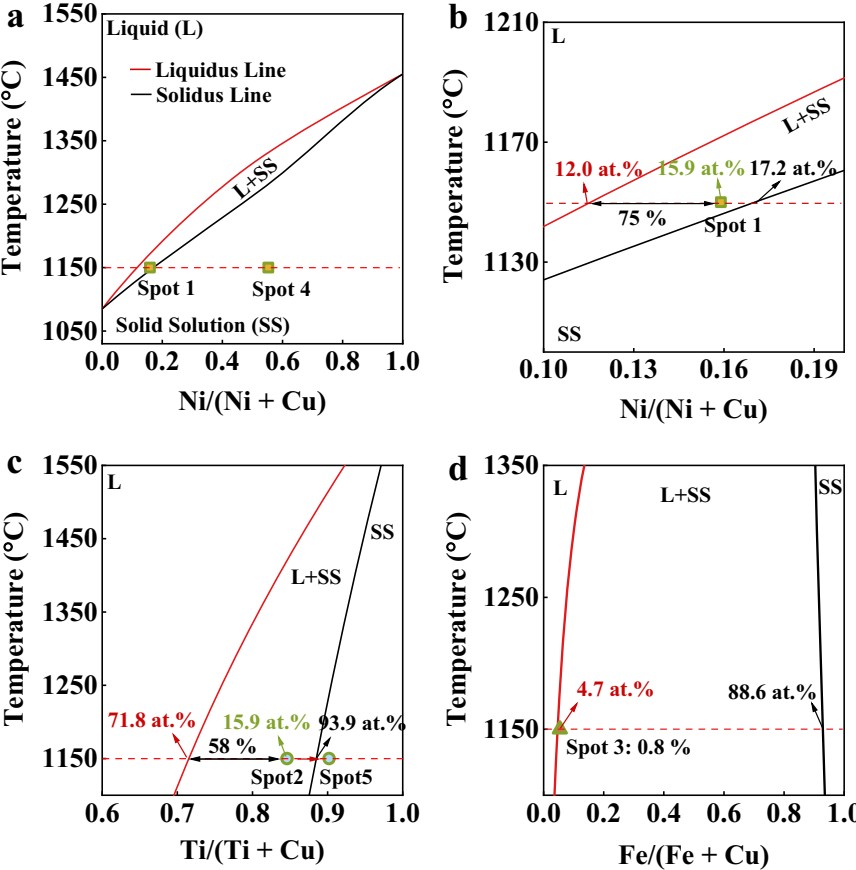

**Fig. 4 | Thermodynamic calculations for different systems.** Equilibrium phase diagrams are calculated for the (**a**, **b**) Cu/Ni, (**c**) Cu/Ti, and (**d**) Cu/Fe systems. The fraction of solids is determined based on the chemical compositions at spots 1–5 marked in Fig. 2 and 3. Detailed calculation procedures for phase diagrams are provided in the Methods section. Source data for (**a**–**d**) are provided as a Source data file.

and defects. To address this, we introduced diatomic solutes (Supplementary Fig. 5) and vacancy defects (Supplementary Fig. 6) into our model. We found that while individual heats of solution vary due to strengthened (Supplementary Fig. 7 and Supplementary Table 1) or weakened (Supplementary Fig. 8 and Supplementary Table 2) Cu–solute bonding, the overall $H_S$ trend (Cu/Ti < Cu/Ni < Cu/Fe) persists. This consistency with experimental observations supports the conclusions derived from our model. Overall, by shedding light on the fundamental mechanism determining the solid solution formation, DFT can be utilized to better understand the wetting behaviors of different metallic systems, as discussed in the following section.

## Spreading dynamics

To validate effects of step flow mechanisms on wetting behaviors, we performed wetting experiments for Cu/Ni, Cu/Ti, and Cu/Fe at 1150 °C using non-contact heating sessile drop methods through a novel Thermal Optical Dynamic Wetting Apparatus (TODWA). The non-contact heating sessile drop method complements contact heating (in-situ observation) by charactering wetting behaviors of the overall droplet from a side view. Figure 6a, b indicate the evolution of contact angles (CAs) and normalized base diameters (BDs), respectively. The normalized BDs is the ratio between the dynamic BD and the initial BD. Cu droplet can rapidly spread on Ti and Fe substrates, reaching equilibrium CAs of 3.5 ± 0.5° and 21.0 ± 0.8° (standard error, three repeated experiments) after 1 s, respectively. In comparison, the spreading of Cu droplets on Ni is relatively slow, and the CA continually decreases to 7.5 ± 0.5° (standard error, three repeated experiments) till 1000 s. The different spreading dynamics in these three systems is further confirmed by the captured droplet profiles at 1 and 100 s (Fig. 6c). This

difference may be attributed to step flow mechanisms rapidly initiating after the contact between Cu and Ni, while it can be only observed for the Cu/Ti system after a long time (2000 s) and does not occur for Cu/Fe. As a result, the step flow mechanism completely controls the spreading of Cu on Ni after the early stage, leading to a slow spreading behavior. With the late presence and absence of the step flow mechanism, the Cu droplets can rapidly spread on Ti and Fe substrates due to the strong metal-metal atom affinity. This is similar to the product forming model, where the lateral growth of IMCs at the L/S interface arrests the CL and results in a slow spreading[24–26]. For instance, Sun et al.[22] reported that the lateral growth of Ti₂Cu leads to a slow spreading for the Cu/Ti system at 1000 °C. Similarly, a comparison of dissolutive Sn/Bi and product forming Sn/Au systems shows fast spreading in Sn/Bi (conform to Case 2) but slower kinetics in Sn/Au at 250 °C due to the IMC 'foot' growth[48,49]. However, spreading in Sn/Au accelerates at 450 °C, where higher Au solubility (81 at. %) delays IMC formation[50]. Thus, early CL arrest from step flow or IMC growth contributes to slow spreading. In contrast to the "pinning−overflow" mode of the step flow mechanism, spreading kinetics in product forming systems is typically linear[24–26]. It should be noted that most steps at the CL are difficult to capture and identify (Fig. 6a) primarily due to the limited resolution of the imaging technique used in the spreading experiments (32 μm/pixel). Only a few steps with a significant CA change of 10° are observed (Supplementary Fig. 9a). Even though the step flow mechanism may be introduced at the late spreading stage in Cu/Ti systems at 1150 °C, it is difficult to track from a side view due to the flattened droplets leading to invalid circle fitting procedures (see droplet profiles in Fig. 6c and fitting errors in Supplementary Fig. 9b). The better final wettability in Cu/Ni and Cu/Ti

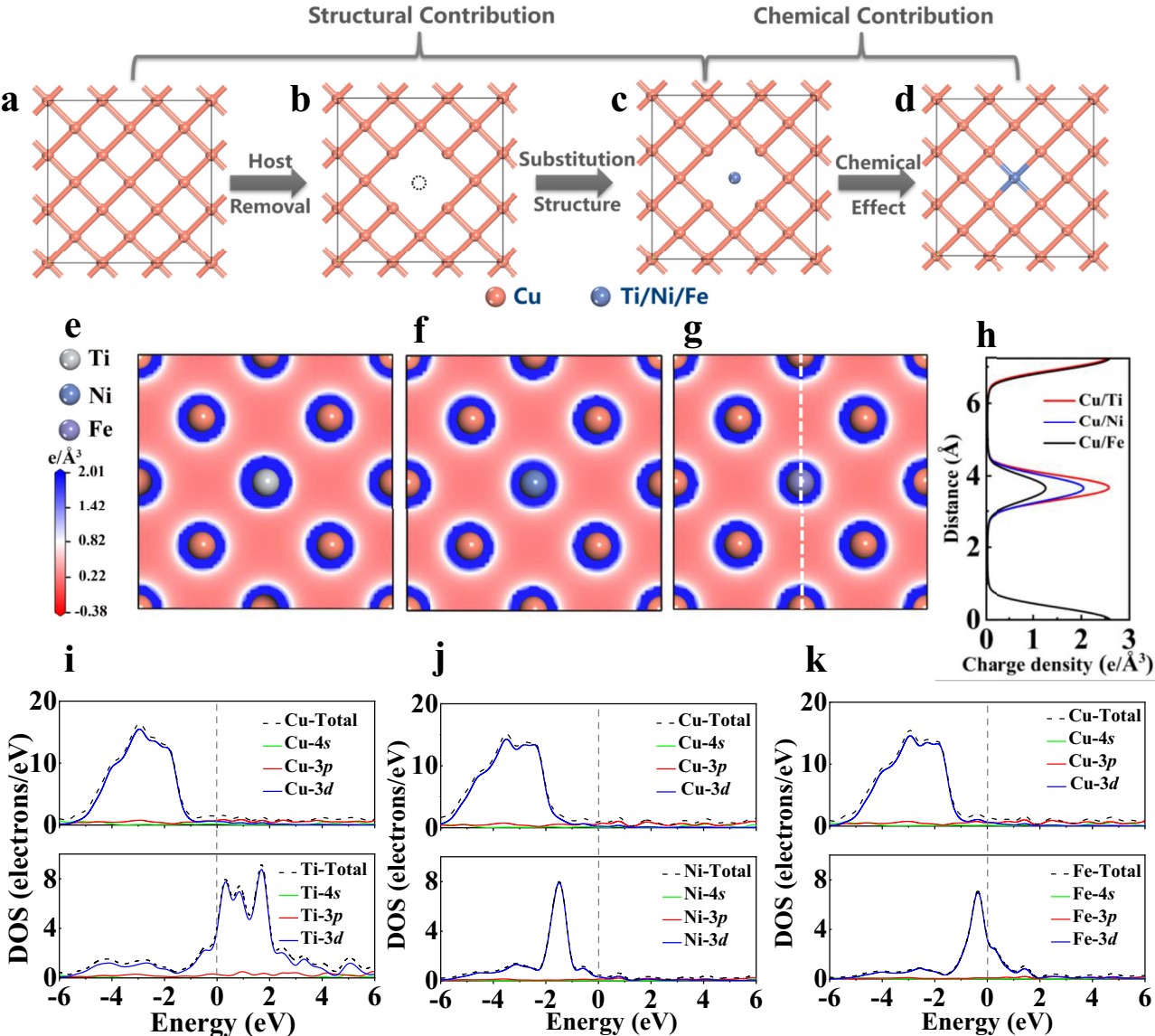

**Fig. 5 | Density functional theory (DFT) calculations. a–d** Schematical illustration of the numerical approach showing the formation of a solvent-solute system starting from a Cu network. From (**a**) to (**b**), formation of an atom vacancy by removing one Cu atom. From (**b**) to (**c**), insertion of a solute atom. From (**c**) to (**d**), formation of chemical bonds between the solvent and solute atom. **e–g** Charge density distribution in (001) plane across different systems. The charge density curve of each system in (**h**) is plotted along the direction of the white dotted line in (**g**). The densities of states of the (**i**) Cu/Ti, (**j**) Cu/Ni and (**k**) Cu/Fe systems reveal different bonding strengths. Source data for (**h–k**) are provided as a Source data file. The atomic coordinates are provided as Supplementary Data 1.

systems compared to that of Cu/Fe systems may be attributed to the different levels of dissolutions. The Ni and even more substantial Ti dissolution into Cu can improve wettability by introducing stronger liquid-solid bonds (as calculated by DFT, Fig. 5), inducing a solutal Marangoni flow towards the CL[23,36], and removing the oxidation film on the substrate surface (if any)[22]. The substrate element dissolution is further evidenced by the BSE pictures of curved L/S interfaces and corresponding EDS analysis near the L/S interface after solidification for Cu/Ni (Fig. 6d, 38.8 at. % Ni), and Cu/Ti (Fig. 6e, 96.5 at. % Ti). At 1150 °C, no IMC layer is observed for Cu/Ti at the L/S interface (vs. Ti$_2$Cu IMC layer at 1000 °C). Instead, only dendritic phases formed during cooling exist, consistent with the phase diagram[51]. The BSE picture of the flat L/S interface (Fig. 6f) and very limited presence of Fe (4.4 at. % Fe) within the Cu droplet demonstrates negligible dissolution. The different levels of substrate elements within the Cu droplets are further confirmed by the EDS mapping of Ni (Fig. 6g), Ti (Fig. 6h) and Fe (Fig. 6i).

## Discussion

We propose the existence of three distinct types of wetting behaviors for dissolutive wetting in metallic systems (Case 1, 2 and 3) depending on the parameters $C_L$ and $C_R$, as illustrated in Fig. 7. The step flow mechanism can rapidly initiates and then controls spreading in systems with a relatively small $C_L$ and $C_R$, leading to a potentially slow spreading dynamic (Case 1). In contrast, it is difficult to introduce step flow behaviors in systems with either a significant $C_L$ or a large $C_R$. In the former, the droplets rapidly spread on metal substrates due to the strong affinity between solid and liquid atoms as evidenced by DFT simulations and step flow mechanism occurs at a later stage after $C_L$ is reached (Case 2). In the latter, no step flow behavior exists and a fast spreading dynamic is expected (Case 3). The proposed mechanism of dissolutive wetting is further confirmed by experimental results from other systems. For instance, the rapid introduction of step flow mechanism in Au/Pt system aligns well with Case 1 (Supplementary Fig. 10 and Supplementary Movie 3). The dissolution behavior and

spreading of Ag on Cu at 1000 °C (Supplementary Fig. 11 and Supplementary Movie 4) and Sn on Cu at 800 °C (Supplementary Fig. 12) conform to Case 2, while those for Ag on Ni at 1150 °C (Supplementary Fig. 13), Sn on Fe at 950 °C (Supplementary Fig. 14 and Supplementary Supplementary Movie 5), and Ag on Ti at 1150 °C (Supplementary Fig. 15) agree with Case 3. In conclusion, we unraveled different wetting behaviors and proposed a thermodynamic approach for assessing wetting properties in dissolutive metallic systems based on $C_L$ and $C_R$. $C_R$ dominates the onset of the step flow mechanism, whereas $C_L$ controls its timing. Through DFT simulations, we provide deeper atomistic insights about the solid solution formation in different cases. For instance, the larger solute compositions in formed solid solutions for Case 1 and 2 compared to Case 3 may result from stronger metallic bonds and enhanced charge transfer. The proposed mechanism offers guidance for optimizing high-temperature processes (e.g., welding, coating, infiltration). For example, during the high-temperature infiltration of Cu–diamond composites, diamond metallization is required to improve wettability with Cu[52]. Ni coatings may improve bonding but slow infiltration, Fe coatings enable fast infiltration but weak adhesion, while Ti coatings may balance both.

**Table 1 | Calculated heat of solution ($H_S$, eV), structural ($H_{SC}$, eV) and chemical ($H_{CC}$, eV) contributions of Ti, Ni, and Fe in Cu crystal structures**

| Solutes | $H_S$ (eV) | $H_{SC}$ (eV) | | $H_{CC}$ (eV) |
|---|---|---|---|---|
| | | $E_{HR}$ | $E_{SS}$ | |
| Ti | 0.03 | 1.38 | 0.16 | −1.51 |
| Ni | 0.28 | 1.38 | 0.02 | −1.12 |
| Fe | 0.65 | 1.38 | 0.01 | −0.74 |

## Methods
Full details are provided in the supplementary materials.

### Materials
The current study utilized highly pure Cu (99.99%, from Chempur company with product number of 7440-50-8), Ag (99.995%, from Chempur company with product number of 7440-22-4), Au (99.99%, from Tehe Metal Materials Co., Hefei, China) and Sn (99.9+ %, from Chempur company with product number of 7440-31-5) as liquid material, and Ni (99.99%, from Chempur company with product number 7440-02-0), Cu (99.999%, from Chempur company with product number of 7440-50-8), Ti (99.995 %, from Chempur company with product number of 7440-32-6), Fe (99.99+ %, from Chempur company with product number of 7439-89-6) and Pt (99.99+ %, from Tehe Metal Materials Co., Hefei, China) as substrate materials. The dimensions of the substrates for the TODWA and CSLM are 16 × 16 × 5 and 5 × 5 × 1 mm (length × width × height), respectively. The substrate surfaces were grounded with silicon carbide papers (180, 300, 800, 1200, 4000 grit) and polished with diamond paste (from 3 to 1 μm). Finally, all substrates were cleaned ultrasonically in ethanol for 30 min, and rinsed by acetone and ethanol.

### In-situ observation and quenching
The Confocal Scanning Laser Microscope (CSLM, Lasertec, 1LM21H-SVF17SP) was used for in-situ observing the spreading of liquid droplets on top of solid substrates and quenching the liquid/solid interfaces[53]. During experiments, purified Ar by passing through a furnace tube filled with Magnesium at 500 °C was utilized, and a low oxygen partial pressure ($P_{O_2}$) of ~3.5 × 10⁻¹⁵ ppm was obtained avoiding oxidation of both liquid droplets and solid substrates according to the Ellingham diagram[54]. Liquid materials on top of solid substrates were

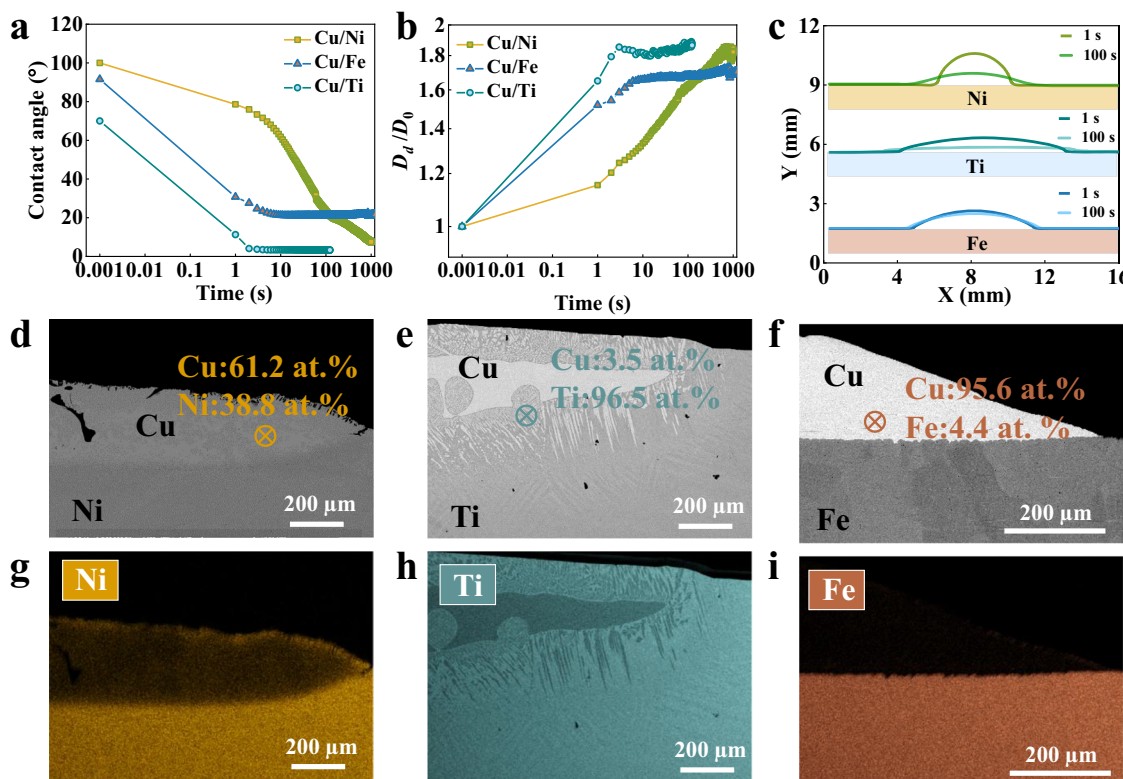

**Fig. 6 | Spreading dynamics and post-solidification microstructure analysis.** Evolution of (**a**) contact angles and (**b**) normalized base diameters for Cu/Ni, Cu/Ti and Cu/Fe. **c** Cu droplet profiles at 1 and 100 s on Ni, Ti, and Fe substrates. Cross-section backscattered electron (BSE) pictures of solidified droplets indicate a curved liquid/solid interface in (**d**) Cu/Ni, (**e**) Cu/Ti, and a flat liquid/solid interface in (**f**) Cu/Fe. The energy dispersive spectroscopy (EDS) mapping pictures reveal (**g**) a significant Ni, (**h**) an even more substantial Ti, but (**i**) a minimal Fe distribution within Cu droplets. Source data for (**a**–**c**) are provided as a Source data file.

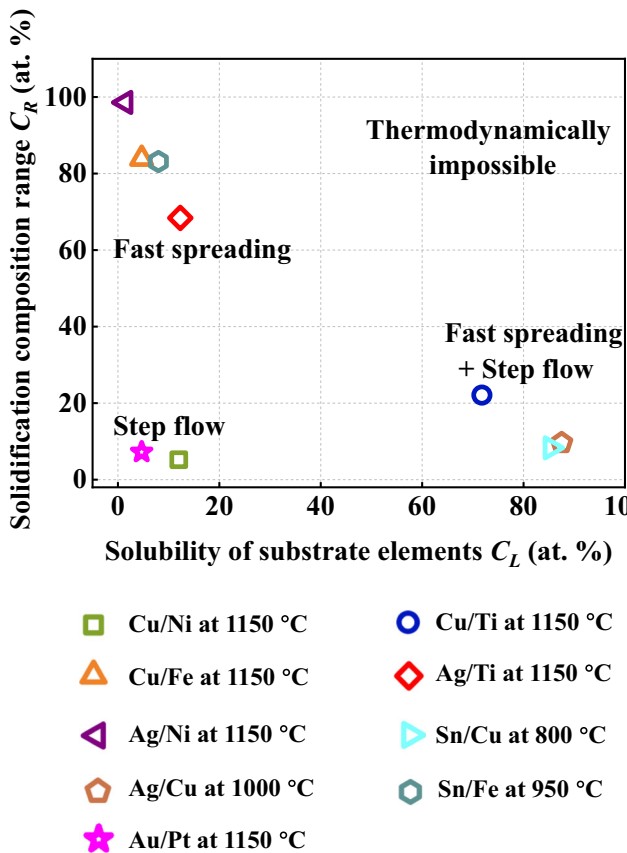

**Fig. 7 | Dissolutive wetting behavior map for metallic systems at test temperatures.** Spreading behaviors in various metallic systems are determined by solubility of substrate elements within liquids ($C_L$) and the solidification composition range ($C_R$). All data originate from this study. Source data are provided as a Source data file.

firstly heated to 180 °C with a heating rate of 50 °C/min, held for 5 min, and then rapidly heated to desired temperatures with a heating rate of 300 °C/min. After 60 min, the liquid droplets together with substrates were quenched by shutting off the furnace power to preserve the liquid/solid interfaces and dissolved substrate elements within droplets. The cooling rate exceeds 500 °C/min[55]. The isothermal stages during experiments were recorded in-situ at a frame rate of 1 fps (frame per second) and with a pixel size of 0.6 μm.

### Non-contact heating sessile drop wetting experiments
We performed wetting experiments by adopting the non-contact heating sessile drop methods through a novel Thermal Optical Dynamic Wetting Apparatus (TODWA, supplementary Fig. 16a). Experiments were performed under purified Ar atmosphere with a $P_{O_2}$ value of ~6.4 × 10$^{-15}$ ppm. The heating/cooling rate during wetting experiments was 3 °C/min and the isothermal stage was 60 min. The temperature control accuracy, heating uniformity, and droplet size effects in the non-contact heating experiments have been detailed in Supplementary Fig. 16b and 17. The whole wetting process was recorded by a high-speed camera (1000 fps with a pixel size of 32 μm) and the wetting properties (e.g., contact angles, base diameters) were extracted from the recorded videos. More detailed information about the wetting process and image processing is given in the supplementary information (Supplementary Fig. 18).

### Microstructure analysis
Samples from both CSLM and TODWA were characterized using a high-resolution scanning electron microscope (SEM XL-30 LaB6)

equipped with an EDS detector (the octane silicon drift detector, SDD). The SE and BSE pictures of the samples were taken from various angles, including top views (with the substrate surface in perpendicular to the electron beam), side views (with the substrate surface in parallel to the electron beam), and cross sections. To obtain the BSE images in cross section, the samples were first cut and prepared by mounting them in epoxy resin, followed by grinding with silicon carbide papers (180, 300, 800, 1200, 4000 grit) and polishing with diamond paste (from 3 to 1 μm). The acceleration voltage and the working distance for the EDS elemental analysis were fixed at 20 kV and -11 mm, respectively.

### Thermodynamic modeling
The theoretical phase diagrams were calculated by the Phase diagram module of FactSage 8.3[56,57]. The calculation employed the FactPS database for pure substances and the FToxid database for oxide phases, with all databases used in their default configurations. The system components were defined as liquid material (Cu, Ag, Au and Sn) solid metals (Ni, Ti, Pt and Fe). For the phase diagram calculation, the "Plot type" was set to "Binary" with the X-axis representing the atomic fraction of solid metals, scanned from 0–100 at. % with a step size of 0.5 at. %. The Y-axis was set to temperature, ranging from 500 °C to 1600 °C. The total pressure was maintained at 1 atmosphere. The $P_{O_2}$ was controlled by explicitly including an $O_2$ gas phase in the equilibrium system within the Phase Diagram module. The final diagram and its underlying numerical data were exported directly from the software for reporting and analysis.

### DFT calculations
A 2 × 2 × 2 supercell of a Cu crystal was constructed, with a single Ti (Ni or Fe) atom substituting for one Cu atom. These calculations were executed using the CASTEP module[58] of the Materials Studio software within the framework of the density functional theory (DFT)[41]. The exchange-correlation functional was represented by the generalized gradient approximation (GGA)[59], specifically through the Perdew–Burke–Engenho (PBE) scheme[60]. To ensure computational accuracy, the Monk horst–Pack scheme was utilized to generate a 7 × 7 × 7 mesh of k-points in the Brillouin zone, and the plane-wave basis set cutoff was set to 450 eV. In addition, the cut-off energy and k-point were tested for convergence (Supplementary Fig. 19). The convergence thresholds were set to 1 × 10$^{-5}$ eV per atom for energy and 0.03 eV/Å for force in the geometry optimization process. We also considered the effects of double solute atoms by simultaneously substituting two Ti/Ni/Fe atoms into the Cu lattice. To assess the defective effects, we introduced vacancy defects by replacing two adjacent Cu atoms with a single solute (Ti, Ni, or Fe) atom.

## Data availability
All relevant data generated in this study are provided in the Supplementary Information or Source data file. The atomic coordinates of the DFT calculation are provided as Supplementary data 1. Source data for images in the main text and Supplementary information are provided with this paper. Source data are provided with this paper.

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

## Acknowledgements

This work was supported by Project supported by the National Science Foundation for Distinguished Young Scholars of China (Grant No. 52501124), China Postdoctoral Science Foundation funded project (Grant No. 2025M770123), Chongqing Natural Science Foundation General Project (Grant No. CSTB2025NSCQ-GPX0561), Major Project of Technological Innovation and Application Development in Chongqing (Grant No. 2024TIAD-STX0069), FWO-Weave project (Grant No. G000225N), KU Leuven internal funding project (Grant No. C14/17/075), internal KU Leuven project (Grant No.AKUL/17). Authors would like to thank Joop Van Deursen and Joris Van Dyck for the technical support on the TODWA construction, Christel Butnaru for the sample preparation, Shu Li for editing supplementary videos and the illustration drawing.

## Author contributions

Y.S. conducted the wetting experiments and in-situ observations, and drafted the initial manuscript. Z.C. carried out the thermodynamic calculations and, together with W.C., assisted in sample preparation. X.C. performed the density functional theory simulations. The study was conceptualized by Y.S., D.S., N.M., and M.G. Supervision of the project was provided by D.S., N.M., Y.S., S.Y. and M.G. The manuscript was reviewed and revised by D.S., N.M., Y.S., S.Y. and M.G. The final version of the manuscript was approved by all authors before submission.

## Competing interests

The authors declare no competing interests.
