## [Transparent Peer Review file · Nature Communications]

Thermodynamic framework for assessing dissolutive wetting behaviors in metallic systems

Corresponding Author: Dr Zhongfu Cheng

Version 0:

Reviewer comments:

Reviewer #1

(Remarks to the Author)

Major Concerns

1. Structural and Logical Issues

Excessive Length in Introduction (Lines 33-76): The background section is overly could be condensed to focus more sharply on the research gap and objectives. Some content may be better suited for the Discussion section.

Blurred Separation Between Results and Discussion: Certain results (e.g., DFT simulations) are intertwined with mechanistic explanations. A clearer distinction between experimental findings and their interpretation is needed.

2. Experimental Methods and Data Presentation

Lack of Critical Experimental Details:

The non-contact heating experiments (TODWA) lack specifics on temperature control accuracy, heating uniformity, and droplet size effects.

The resolution difference between CSLM (0.6 $\mu\text{m}/\text{pixel}$) and TODWA (32 $\mu\text{m}/\text{pixel}$) is significant but not discussed in terms of its impact on step-flow observation sensitivity.

Insufficient Statistical Analysis:

Error margins for contact angles (e.g., " $\pm 0.5^\circ$ ") are provided, but the number of replicates and statistical methods (e.g., standard deviation vs. confidence intervals) are unclear.

Only one quenched sample per system (Cu/Ni, Cu/Ti, Cu/Fe) is shown, raising concerns about reproducibility.

3. Theoretical Modeling and Computations

Limitations of DFT Simulations:

The substitutional solute model (single Ti/Ni/Fe atom in Cu lattice) oversimplifies real interfacial interactions, neglecting multi-atom effects, defects, or grain boundaries.

Convergence tests for DFT parameters (k-point mesh, cutoff energy) are absent, casting doubt on the reliability of calculated heats of solution (H_s).

Generalizability of C_s and C_r : The claim that these parameters universally predict wetting behavior should be tempered, as non-ideal systems (e.g., short-range ordered alloys) may deviate.

4. Discussion and Conclusions

Mechanistic Evidence Lacking:

The proposed step-flow mechanism relies on indirect observations (e.g., sharp edges in quenched samples). Direct evidence (e.g., in situ composition/velocity measurements at the contact line) would strengthen the argument.

The assertion that high C_r (83.9 at% for Cu/Fe) prevents step-flow is qualitative; a quantitative threshold or kinetic argument is needed.

Weak Link to Applications: While the conclusion mentions implications for welding/coating, no concrete examples or guidelines for process optimization are provided.

5. Formatting and Clarity Issues

Figure/Tables Need Improvement:

Figure 1b's schematic labels are too small to read.

Table 1's symbols (e.g., " $H_s(\text{eV})$ ") should follow consistent formatting (italic/roman).

Term consistencies: "Step flow" is alternately called "step-wise spreading"; standardize usage.

Redundant References: Lines 532–533 cite the same study twice with nearly identical text.

6 . The authors focus on purely dissolutive wetting, yet the systems they studied are not purely dissolutive. The selected representative systems involve both dissolution and the saturation precipitation of intermetallic compounds (IMCs), and the influence of the latter cannot be entirely ruled out.

For purely dissolutive wetting, the establishment of a thermodynamic framework should generally be based on the following two aspects:

First, the changes in solid/liquid interface geometry caused by dissolution must be considered.

Second, the effect of dissolved solutes (whether in the liquid phase or as solid solutions) on interfacial tension must be accounted for.

These key points were not adequately addressed by the authors. To properly study purely dissolutive wetting, the authors should focus on the following

[1] L. Yin, Reactive wetting and spreading in binary metallic systems, Philosophy in Materials Engineering in the Graduate School, Binghamton University, New York, 2005, p. 188.

[2] L. Yin, A. Chauhan, T.J. Singler, Reactive wetting in metal/metal systems: Dissolutive versus compound-forming systems, Materials Science and Engineering: A 495(1-2) (2008) 80-89.

[3] L. Yin, et al., Reactive wetting in metal–metal systems, Journal of Physics: Condensed Matter 21(46) (2009) 464130.

[4] L. Yin, S.J. Meschter, T.J. Singler, Wetting in the Au-Sn System, Acta Materialia 52(10) (2004) 2873-2888.

[5] L. Yin, B.T. Murray, T.J. Singler, Dissolutive wetting in the Bi-Sn system, Acta Materialia 54(13) (2006) 3561-3574.

[6] L. Yin, B.T. Murray, S. Su, S. Ying, T.J. Singler, Reactive wetting in metal–metal systems, J Phys Condens Matter 21(46) (2009) 464130.

[7] J.A. Warren, W.J. Boettinger, A.R. Roosen, Modeling reactive wetting, Acta Materialia 46(9) (1998) 3247-3264.

[8] R. Warren, Solid-liquid interfacial energies in binary and pseudo-binary systems, Journal of Materials Science 15(10) (1980) 2489-2496.

Reviewer #2

(Remarks to the Author)

The authors present a comprehensive study of dissolutive wetting in metallic systems. Two thermodynamic factors have been identified to control the spreading kinetics. The variety of experimental and modeling techniques have been used to deliver the complete picture of the process, and enough details have been provided to reproduce the research. The manuscript is well written and only minor issues have been found such as some digits introduced in the abstract and sentence grammar on page 10 line 189 should be fixed. Overall, the work represent significant advancement to the field, and can be accepted after minor revision.

Reviewer #3

(Remarks to the Author)

The authors combine experiments, thermodynamic calculations, and density functional theory based atomistic simulations to assess thermodynamic driving forces in dissolutive wetting systems. They convincingly show that the interplay between wetting, spreading, and inter-metallic compound formation can be predicted based on two quantities: the substrate solubility in the droplet liquid (C_S) and the composition range for solidification (C_R). The paper is well written, and the figures and tables support well the conclusions drawn. I particularly enjoyed the detailed view provided by many of the figures in both the manuscript and the supplementary material. I provide mostly minor suggestions below and one suggestion that is less trivial. I save the less trivial one for last, so it is not presented in its order of appearance in the manuscript.

1) The flat curve for Cu/Fe in Fig. 1 is initially confusing, because fast spreading is expected; perhaps the observation that spreading is already done by the start of the isothermal stage should be made a bit earlier in the text (e.g., when the figure is first introduced). In general, at that point, it would be useful to briefly describe any spreading behavior observed during heat up.

2) In the Fig. 1 caption "evidences" should be "evidence".

3) Lines 185-186: The sentence, " CR accounting for an interval with coexisting liquid and solid phases [38].", is not clear.

4) Line 189: "solidifies" should be "solidify".

5) Line 256: put "contribution" after "chemical".

6) Line 257: The labels H_{SC} and H_{CC} are reversed.

7) Line 274: "session" should be "section".

8) Lines 385-386: the sentence, "The cooling rate during exceeds 500 °C/min [50].", needs fixing.

9) In Fig. 6 (and in the supplementary material) the authors present data for other chemical systems (i.e., in addition to Cu/Ni, Cu/Ti, and Cu/Fe). Those data help bolster the arguments made in the manuscript using the 3 primary test cases (Cu/Ni,

Cu/Ti, and Cu/Fe). For each of the the latter 2 of those cases, data for at least 2 other similar chemical systems (as far as C_S and C_R) are presented in Fig. 6. For the Cu/Ni case, no additional supporting chemical systems are explored. To be clear: the authors have done a lot of very elegant work here; nonetheless, it would be compelling if one more case of low C_R and low C_S were shown to exhibit behavior similar to Cu/Ni. The authors compare this system to what I think of as reaction limited reactive wetting systems, and many such systems have been presented in prior literature. So perhaps the authors don't feel it is necessary to show an additional system. But perhaps they could at least talk about existing literature for comparable systems and show that analogous behavior is exhibited?

Version 1:

Reviewer comments:

Reviewer #1

(Remarks to the Author)

The author provided a detailed response and made corresponding revisions in the revised manuscript. Ultimately, I believe that the paper is acceptable and should be published.

Reviewer #2

(Remarks to the Author)

The authors substantially improved their manuscript, which should be ready for publication now.

Reviewer #3

(Remarks to the Author)

The authors have fully addressed any concerns I raised about their initial manuscript submission, and I believe it is ready for publication.

**Thermodynamic framework for assessing dissolutive wetting**
**behaviors in metallic systems**

Youqing Sun ^{1,2,3}, Shoufeng Yang^{1,2*}, Xinghong Cai ^{3,4*}, Zhongfu Cheng^{3*}, Wantong Chen³, Nele
Moelans³, Muxing Guo³, David Seveno³

**Affiliations**

**1** *Chongqing Institute of Green and Intelligent Technology, Chinese Academy of*
*Sciences, Chongqing 400714, China*

Author-1, Author-2

**2** *University of Chinese Academy of Sciences, Chongqing School, Chongqing 400714,*
*PR China*

Author-1, Author-2

**3** *Department of Materials Engineering, KU Leuven, Kasteelpark Arenberg 44, 3001*
*Leuven, Belgium*

Author-1, Author-3, Author-4, Author-5, Author-6, Author-7 & Author-8

**4** *Chongqing Key Laboratory for Advanced Materials and Technologies of Clean*
*Energies, School of Materials and Energy, Southwest University, Chongqing 400715,*
*China*

Author-3

Corresponding author

Correspondence to: Author-2; Author-3; Author-4

Dear Editor,

Dear esteemed reviewers,

We express our sincere gratitude for your invaluable feedback on our manuscript titled
'**Thermodynamic framework for assessing dissolutive wetting behaviors in metallic systems.**'

Your constructive and insightful comments have been instrumental in guiding us towards enhancing

the quality of our work. We deeply appreciate the time and effort you dedicated to reviewing our
draft.

Your thoughtful remarks have inspired meaningful revisions, and we have diligently incorporated
all your suggestions into our manuscript. We have carefully considered each comment, and the
corresponding modifications have been integrated into the text and we provided answers to the
questions you raised to the best of our abilities. The primary changes are as follows:

● We strengthened the step flow mechanism by incorporating experimental and simulation
results, and by comparing equilibrium Cu–10 at. %Ni / Cu–30 at. %Ni with the non-
equilibrium Cu/Ni system. **(Responses 10, 18, 29)**

● We validated the proposed thermodynamic framework by conducting Au/Pt experiments (low
C_L and low C_R regime). **(Responses 18, 29)**

● We clarified the different liquid/solid interactions and wetting behaviors in product forming
versus dissolutive systems by comparing Cu/Ti and Sn/Fe at different temperatures.
**(Responses 17, 18)**

● We validated our atomistic model by involving multi-atom and defect effects. **(Response 7)**

For your convenience, we have outlined our responses to your comments in green color, with the
associated changes highlighted in the revised manuscript in light blue color in this letter.

We are eager to address any additional feedback you may offer to further refine our work.

Thank you once again for your time and expertise.

Best regards,

Zhongfu Cheng

**Reviewer #1 (Remarks to the Author):**

Major Concerns

1. Structural and Logical Issues

**Comment 1:** Excessive Length in Introduction (Lines 33-76): The background section is overly
could be condensed to focus more sharply on the research gap and objectives. Some content may be
better suited for the Discussion section.

**Response 1:** Thank you for this valuable suggestion. We have revised the introductory part

accordingly to better highlight the research gap and objectives. The introduction is now more
concise, reduced from lines 33–76 to lines 33–64:

● We have removed the detailed explanation of the step flow mechanism from the introduction,
as this is already thoroughly discussed in the Results section.

**Related revisions 1:** It involves (i) pinning of the CL by the solidified edge due to solid
solution formation at the L/S interface, (ii) Marangoni-driven liquid flow towards the edge,
(iii) precursor film formation from enhanced liquid–solid affinity, and (iv) sudden overflow
due to liquid accumulation at the edge. **(Lines 112-117 in the main text)**

● We have moved some parts to the Discussion section.

**Related revisions 1:** This is similar to the product forming model, where the lateral growth of
IMCs at the L/S interface arrests the CL and results in a slow spreading^{1–3}. For instance, Sun
et al.⁴ reported that the lateral growth of Ti₂Cu leads to a slow spreading for the Cu/Ti system
at 1000 °C. Similarly, a comparison of dissolutive Sn/Bi and product forming Sn/Au systems
shows fast spreading in Sn/Bi (conform to Case 2) but slower kinetics in Sn/Au at 250 °C due
to IMC ‘foot’ growth^{5,6}. However, spreading in Sn/Au accelerates at 450 °C, where higher Au
solubility (81 at. %) delays IMC formation⁷. Thus, early CL arrest from step flow or IMC
growth contributes to slow spreading. In contrast to the "pinning–overflow" mode of the step
flow mechanism, spreading kinetics in product forming systems is typically linear^{1–3}. **(Lines**
**309-318 in the main text)**

**Comment 2:** Blurred Separation Between Results and Discussion: Certain results (e.g., DFT
simulations) are intertwined with mechanistic explanations. A clearer distinction between
experimental findings and their interpretation is needed.

**Response 2:** Thank you for your suggestion. We have better separated the DFT result section to the
Discussion section.

**Related revisions 2:**

**Results:** Table 1 indicates H_S for different systems. To further decipher the chemical contribution,
planar charge density distributions were calculated for Cu/Ti (Figure 4e), Cu/Ni (Figure 4f), and
Cu/Fe (Figure 4g). The local electron density near each solute atom is compared in Figure 4h.
Additionally, density of states (DOS) near the Fermi level (Figure 4i–k) were extracted for all

systems, showing overlapping Cu-3d and solute-3d orbitals in the energy range of -5.0 to -2.0 eV.
The magnitude of the DOS peak in this energy range follows Ti > Ni > Fe. **(Lines 250-256 in the**
**main text)**

**Discussions:** DFT results indicate that Cu atoms are more easily substituted by Ti than by Ni or Fe,
as confirmed by the order of H_S : Cu/Ti (0.03 eV) < Cu/Ni (0.28 eV) < Cu/Fe (0.65 eV).
Consequently, the Cu crystal can accommodate more Ti atoms than Ni, and Fe atoms can hardly
replace Cu to form solid solutions. This trend is confirmed by experimental observations, showing
that the Ti concentration (spot 5 in Figure 2g) is higher than the Ni concentration (spot 4 in Figure
2e) in solid solutions, while only a limited Fe concentration (spot 3 in Figure 2b) is observed in the
solid solution in the Cu/Fe systems. This large difference is primarily due to the chemical
contribution (H_{CC}) but not the structural contribution (H_{SC}) according to Table 1. In other words, the
bonding between the solute atom and neighboring host Cu atoms governs substitution energetics
more than the lattice distortion. The higher charge density in the vicinity of the Ti atom than that of
the Ni and Fe atom indicates a stronger Cu-Ti bond compared to the Cu-Ni and Cu-Fe bonds. The
order of the magnitude of the DOS peak characterizes the chemical bond strength formed by orbital
hybridization, i.e. Cu-Ti > Cu-Ni > Cu-Fe. The higher Ti concentration in Cu-Ti solid solutions is
then due to stronger metallic bonds with Ti-3d orbitals significantly contributing to the DOS, leading
to an improved conductivity and enhancing charge transport. We acknowledge that our
substitutional solute model ignores factors such as multi solute atoms and defects. To address this,
we introduced diatomic solutes (Figure S6) and vacancy defects (Figure S7) into our model. We
found that while individual heats of solution vary due to strengthened (Figure S8 and Table S1) or
weakened (Figure S9 and Table S2) Cu–solute bonding, the overall trend (Cu/Ti < Cu/Ni < Cu/Fe)
persists. This consistency with experimental observations supports the conclusions derived from our
model. Overall, by shedding light on the fundamental mechanism determining the solid solution
formation, DFT can be utilized to better understand the wetting behaviors of different metallic
systems, as discussed in the following section. **(Lines 257-281 in the main text)**

2. Experimental Methods and Data Presentation

Lack of Critical Experimental Details:

**Comment 3:** The non-contact heating experiments (TODWA) lack specifics on temperature control

accuracy, heating uniformity, and droplet size effects.

**Response 3:** Thank you for your suggestions, we have clarified details about temperature control
accuracy, heating uniformity, and droplet size effect in non-contact heating experiments as follows:

**(1) Temperature control accuracy and heating uniformity:**

We calibrated the temperature distribution inside the Al₂O₃ tube using two thermocouples (Figure
A1a): one positioned near the estimated location above the droplet (TC1), and the other at the bottom
of the inner horizontal tube (TC2). The small temperature difference between them (shown in Figure
A1b) confirms that the target temperatures for the liquid droplets are accurate. Details regarding
temperature calibration, as well as temperature control parameters (e.g., heating rates, cooling rates,
and isothermal holding stages), are now included in the Supplementary Material (SM):

**Related revisions 3:**

**Main text:** The temperature control accuracy, heating uniformity, and droplet size effects in the
non-contact heating experiments have been detailed in the SM (Figure A1 and A2). **(Lines 412-414**
**in the main text)**

**Supplementary materials:** Before starting experiments, the temperature difference inside the
Al₂O₃ tubes is calibrated by comparing the measurements from two thermocouples (Figure A1a),
one is located at a position above the droplet (TC1), while another is placed at the bottom of the
inner horizontal tube (TC2). The small difference between them indicates that the temperatures
designed for our liquid droplets are accurate (Figure A1b). The solid Cu material is firstly loaded
into an Al₂O₃ nozzle mounted on an injector, and then heated separately with the substrate at a rate
of 3 °C/min. When it reaches the desired temperatures, the liquid Cu is squeezed out by a plunger
and comes into contact with the substrate. LED light passes through the window, and shadow images
of both the Cu droplet and Ni surface are captured by a high-speed camera (Optronis CP70-1-M/C-
1000). Purified Ar is flushed through the Al₂O₃ tubes during the whole process. The oxygen partial
pressure inside the chamber is monitored in-situ by an oxygen sensor (15, Rapidox 2100 Gas
Analyser). The isothermal stage lasts 1 hour, the furnace then starts to cool down at a rate of 3
139 °C/min. **(Lines 34-45 in the SM)**

[Figure Redacted]

**Figure A1.** (a) Schematic of the experimental setup, TODWA. (b) Temperature difference between
TC1 and TC2.

**(2) Size effects:**

We do agree that large drops may influence wetting as a result of increased gravitational effects
compared to capillary forces. but we excluded such size effects by calculating the Bond number
(B_0). The main text and supplementary materials have been revised accordingly.

**Related revisions 3:**

**Main text:** The temperature control accuracy, heating uniformity, and droplet size effects in the
non-contact heating experiments have been detailed in the SM (Figure A1 and A2). **(Lines 412-414**
**in the main text)**

**Supplementary materials:** The Bond number (B_0) balances out the effects of gravity and surface
tension, and can be used to characterize droplet size effects (Equation A2).

$$B_0 = \frac{\Delta\rho g R^2}{\sigma_{LV}} \quad (\text{A2})$$

where $\Delta\rho$ is the difference between the liquid Cu density and that of the surrounded atmosphere.
 $\Delta\rho$ can be considered to be equal to the density of liquid Cu (7850 kg/m^3 at $1150\text{ }^\circ\text{C}$ ⁸). g is the
 gravitational acceleration (9.8 m/s^2). R is the droplet radius of the droplet (see Figure A2a). σ_{LV}
 is the surface tension of liquid Cu taken as 1.3 N/m at $1150\text{ }^\circ\text{C}$ ⁹.
 When $B_0 < 1$, capillary effects dominate over gravity. The droplets tend to adopt a 3D spherical
 cap shape with a constant curvature (a circle in 2D). The contact angles derived from circle fitting
 are then valid. In contrast, when $B_0 > 1$, significant droplet flattening occurs due to gravity. Figure
 A2a and b show the droplet radius and B_0 for the two Cu/Ni samples at $1150\text{ }^\circ\text{C}$ during non-contact
 heating experiments. Ensuring a consistent droplet size is challenging due to the small volume and
 mass of droplets (Figure A2c). $B_0 < 1$ is found for both cases, confirming the minimal effects of
 droplet size. Small droplets (typically $0.01\text{--}0.1\text{ g}$) are preferred for accurate contact angle
 measurements, as gravity-induced deformation is minimal¹⁰. Larger droplets ($>1\text{ g}$) are required
 only when measuring surface tension¹⁰. In our case, the droplet mass is 0.13 g , so size effects can
 be safely neglected. **(Lines 68-86 in the SM)**

 **Figure A2.** (a) Droplet radius for Cu/Ni systems. (b) calculated Bond number and (c) droplet
 volume.

**Comment 4:** The resolution difference between CSLM (0.6 $\mu\text{m}/\text{pixel}$) and TODWA (32 $\mu\text{m}/\text{pixel}$)
is significant but not discussed in terms of its impact on step-flow observation sensitivity.

**Response 4:** Thank you for this comment. Indeed, the resolution difference between CSLM
(0.6 $\mu\text{m}/\text{pixel}$) and TODWA (32 $\mu\text{m}/\text{pixel}$) affects the sensitivity of the step-flow detection. We have
previously addressed this point in our earlier publication (Sun et al. Acta Materialia 282 (2025):
120519)¹¹.

In this paper, Figure A3a shows two CL steps of a Cu droplet captured by the TODWA: a large step
of $\sim 250 \mu\text{m}$ (from point α to β) and a smaller step of $\sim 100 \mu\text{m}$ (from β to γ). The extracted droplet
profiles (Figure A3b) confirm that these steps occur rapidly (within 1 s). No steps larger than 100 μm
were observed beyond these events. However, we acknowledge that smaller steps ($< 100 \mu\text{m}$) may
still occur but are not observable due to the limited resolution of the TODWA. The 32 $\mu\text{m}/\text{pixel}$
scale is insufficient to reliably capture fine step flow motion at the CL.

In contrast, CSLM, with its higher spatial resolution, enables in situ tracking of smaller steps. As
shown in Figure A3c, images obtained by the CSLM captures the detailed progression of the CL in
the Cu/Ni system at 1150 $^{\circ}\text{C}$. Figure A3d illustrates one such step-flow event: the CL remains pinned
from point 1 to point 2, then suddenly advances by $\sim 200 \mu\text{m}$ to point 3, followed by another pinning
stage at point 4. It can be seen that steps even smaller than 20 μm (Figure A3c) can be clearly
resolved by CSLM due to its much smaller pixel size (0.6 $\mu\text{m}/\text{pixel}$).

**Figure A3.** The step flow behavior is observed from (a) wetting experiments at the macro scale,
and the droplet profiles in (b) indicate the CL instantaneously advances in (a). (c) the step flow

behavior from in-situ observation at 1100 and 1150 °C. (d) the microscopic pictures with in detail
one representative process of step flow indicated as 1, 2, 3 and 4 in (c).

We have clarified the impact of resolution differences on detecting step flow mechanisms and
explained the rationale for using different experimental methods in both the revised main text and
SM:

**Related revisions 4:**

**Main text:** The non-contact heating sessile drop method complements contact heating by
characterizing wetting behavior of the overall droplet from a side view. **(Lines 92-95 in the main
text)**

It should be noted that most steps at the CL are difficult to capture and identify (Figure 5a) primarily
due to the limited resolution of the imaging technique used in the spreading experiments (32
$\mu\text{m}/\text{pixel}$). Only a few steps with a significant CA change of 10° are observed (Supplementary
Figure S10a). **(Lines 318-321 in the main text)**

**Supplementary material:** The non-contact heating sessile drop method performed with the
Thermal Optical Dynamic Wetting Apparatus (TODWA) during wetting experiments can capture
the spreading at the initial stage (1000 images per second (s)), and characterize the spreading
behavior of the overall droplet from a side view. However, the samples cannot be quenched to
evaluate the dissolution and solidification during the isothermal stage due to the use of a thermal-
shock sensitive alumina T-junction. To overcome this difficulty, we performed in-situ observation
(top view) by conducting contact heating sessile drop method and the subsequent quenching in the
Confocal Laser Scanning Microscope (CSLM) with a cooling rate larger than $500^\circ\text{C}/\text{min}$. The top-
view observation, especially focusing on the movement of the CL, help complementing our
interpretation of the spreading process. **(Lines 13-22 in the SM)**

Insufficient Statistical Analysis:

**Comment 5:** Error margins for contact angles (e.g., " $\pm 0.5^\circ$ ") are provided, but the number of
replicates and statistical methods (e.g., standard deviation vs. confidence intervals) are unclear.

**Response 5:** Thank you for pointing this out, we have revised our description in the main text.

**Related revisions 5:** We can see that the Cu droplet can rapidly spread on Ti and Fe substrates,
reaching equilibrium CAs of $3.5 \pm 0.5^\circ$ and $21.0 \pm 0.8^\circ$ (standard error, 3 repeated experiments)
after 1 s, respectively. In comparison, the spreading of Cu droplets on Ni is relatively slow, and the

CA continually decreases to $7.5 \pm 0.5^\circ$ (standard error, 3 repeated experiments) till 1000 s. **(Lines**
**297-301 in the main text)**

**Comment 6:** Only one quenched sample per system (Cu/Ni, Cu/Ti, Cu/Fe) is shown, raising
concerns about reproducibility.

**Response 6:** Thank you for your question. Reproducibility is indeed an important point. We
approached it from two aspects:

**(1) Reproducibility per system**

Three repeats were performed for each Cu/Ni, Cu/Ti, and Cu/Fe system. It is now clarified in the
revised paper.

**Related revisions 6:**

**Main text:**

It is interesting to note that the step flow behavior initiates early after the beginning of the isothermal
stage for the Cu/Ni system, while it emerges after ~ 2000 s for the Cu/Ti system (experiments are
repeated three times as shown in Figure A4a in the SM). **(Lines 123-126 in the main text)**

The microstructure of repeated Cu/Ni experiments is detailed in Figure A4b-e. **(Lines 142 in the**
**main text)**

**Supplementary material:**

Figure A4a indicates that the step flow mechanism initiated rapidly after the beginning of the
isothermal stage for Cu/Ni systems, while it emerges after ~ 2000 s for the Cu/Ti systems. No step
flow mechanism is observed for Cu/Fe systems. Repeated EDS analyses of Cu/Ni (Figure A4b-e)
confirm reproducibility: quenched droplets consistently show good wetting with sharp edges, where
Ni concentration is ~ 55 at. % at the edge and ~ 15 at. % away. **(Lines 87-95 in the SM)**

Figure A4. (a) Repeated experiments for Cu/Ni, Cu/Ti and Cu/Fe systems. (b-e) Microstructure characterization and EDS analysis for quenched Cu samples.

**(2) Reproducibility of low C_L and low C_R regime**

We also explored the additional Au/Pt system to demonstrate reproducibility particularly in the low
C_L and low C_R regime populated only Cu/Ni in the first version of this paper. Figure A5a shows
the CL advancement during the isothermal stage (contact heating) for an Au/Pt system ($C_L = 4.7$
at. % and $C_R = 7.1$ at. %) at 1150 °C. The step-flow mechanism is observed from the beginning of
the isothermal stage, with one representative step occurring between 1723–1789 s. During this
interval, the CL remains pinned at the solidified edge from 1723 to 1788 s, before being released at
1789 s as liquid Au overflows the edge due to Marangoni flow and liquid accumulation at the CL
(Figure A5b). The Au/Pt system was quenched at 3600 s. The flattened droplet after quenching

indicates good wetting between liquid Au and solid Pt (Figure A5c), with a sharp edge clearly
observed at the CL (Figure A5d). EDS mapping reveals that the solidified edge is Pt-rich (Figure
A5e). EDS Point analysis further shows a Pt concentration of 25.2 at. % at the edge, which decreases
to 8.6 at. % away from the edge (Figure A5d and g). In addition to the top view, the sharp edge is
also confirmed in the side view (Figure A5f). From a thermodynamic perspective, the quenched Pt
composition at spot S1 confirms that the sharp edge results from isothermal solidification, while
regions of the droplet far from the solidified edge remain liquid during the isothermal stage (spot
S2 in Figure A5h).

In summary, the Au/Pt system represents a case similar to the Cu/Ni system, both characterized by
low C_R and low C_L . The spreading mechanism is identical in both systems: C_R controls the onset
of the step-flow mechanism, while C_L determines its timing.

**Related revisions 6:**

**Main text:**

These 9 systems (Cu/Ti, Cu/Ni, Cu/Fe, Au/Pt, Ag/Ni and Ag/Ti at 1150 °C, Sn/Cu at 800 °C, Ag/Cu
at 1000 °C and Sn/Fe at 950 °C) can be categorized into three cases according to the combination
of two thermodynamic parameters: the solubility of the substrate metal within the liquid metal (C_L)
and the solidification composition range (C_R) between the liquidus and solidus line at a given
temperature. **(Lines 72-76 in the main text)**

The proposed mechanism of dissolutive wetting is further confirmed by experimental results from
other systems. For instance, the rapid introduction of step flow mechanism in Au/Pt system aligns
well with Case 1 (Supplementary Figure A5 and Video S3). **(Lines 356-359 in the main text)**

**Supplementary material:**

**Figure A5.** (a) Travelling distance of the CL vs time illustrating the step flow mechanism of Au on

Pt from the beginning of the isothermal stage at 1150 °C, as confirmed by in-situ observation in (b).

The SEM picture of quenched sample shows (c) a flattened Au droplet and (d) the existence of the

sharp edge at the contact line. The EDS mapping indicates (d) Pt distribution within Au droplets.

The sharp edge is further confirmed by the (f) side view image. The Pt compositions at the positions

near the contact line and at the sharp edge are shown in (g). Corresponding phase diagram in (h)

indicates that the sharp edge results from isothermal solidification, while regions of the droplet far

from the solidified edge remain liquid during the isothermal stage. **(Lines 139-149 in the SM)**

3. Theoretical Modeling and Computations

Limitations of DFT Simulations:

**Comment 7:** The substitutional solute model (single Ti/Ni/Fe atom in Cu lattice) oversimplifies
real interfacial interactions, neglecting multi-atom effects, defects, or grain boundaries.

**Response 7:** Thank you for your comment. The theoretical calculations in this study aimed to obtain
the heat of solution and electronic interactions between solute atoms and the Cu matrix. Using a
perfect substitutional model eliminates interference from grain boundaries and other complex
factors, thereby providing fundamental insights into solute–Cu bonding. We agree that multi-atom
effects and defects may also influence interfacial interactions. To address this, we have examined
effects of diatomic solutes (Figure A6) and vacancy defects (Figure A7). The heat of solution is
lower for diatomic systems primarily due to enhanced chemical bonding between solute atoms and
the Cu matrix through d-d orbital hybridization (Figure A8 and Table A1). In contrast, vacancy-
containing systems show higher values because defects weaken solute–Cu interactions (Figure A9
and Table A2). Interestingly, all cases preserve the same trend ($\text{Cu/Ti} < \text{Cu/Ni} < \text{Cu/Fe}$), confirming
the reliability of our approach. We acknowledge that modeling grain boundaries requires supercells
beyond conventional DFT capabilities and will be pursued in future studies. Discussions on the
effects of diatomic solutes and defects have been added in the revised paper, with more systematic
studies planned for a forthcoming work.

**Related revisions 7:**

**Main text:** We acknowledge that our substitutional solute model ignores factors such as multi solute
atoms and defects. To address this, we introduced diatomic solutes (Figure A6) and vacancy defects
(Figure A7) into our model. We found that while individual heats of solution vary due to
strengthened or weakened Cu–solute bonding (Figure A8 and 9), the overall trend ($\text{Cu/Ti} < \text{Cu/Ni}$
$< \text{Cu/Fe}$) persists (Table A1 and 2). This consistency with experimental observations supports the
conclusions derived from our model. **(Lines 273-279 in the main text)**

We also considered the effects of double solute atoms by simultaneously substituting two Ti/Ni/Fe
atoms into the Cu lattice. To assess the defective effects, we introduced vacancy defects by replacing
two adjacent Cu atoms with a single solute (Ti, Ni, or Fe) atom. **(Lines 445-448 in the main text)**

**Supplementary materials:**

To simulate the effects of multi-atoms and defects on atomic interactions, we introduced diatomic
 solutes (Figure A6) and vacancy defects (Figure A7) into our model. We found that while individual
 heats of solution vary due to strengthened or weakened Cu–solute bonding (Figure A8 and 9), the
 overall trend ($\text{Cu/Ti} < \text{Cu/Ni} < \text{Cu/Fe}$) persists (Table A1 and 2). This consistency with experimental
 observations supports the conclusions derived from our model. (Lines 112-133 in the SM)

 **Figure A6.** Schematic illustration of the numerical approach showing the formation of a solvent-
 solute system (diatomic system) starting from a Cu network.

 **Figure A7.** Schematic illustration of the numerical approach showing the formation of a solvent-
 solute system starting from the Cu network with vacancy defects.

 **Figure A8.** The densities of states of (a) Cu/Ti, (b) Cu/Ni and (c) Cu/Fe for the diatomic system.

Figure A9. The densities of states of (a) Cu/Ti, (b) Cu/Ni and (c) Cu/Fe for the defect system.

Table A1. Calculated heat of solution (H_S , eV), structural (H_{SC} , eV) and chemical (H_{CC} , eV) contributions of Ti, Ni, and Fe in Cu crystal structures (diatomic system).

Solutes	H_S (eV)	H_{SC} (eV)		H_{CC} (eV)
		E_{HR}	E_{SS}	
Ti	-0.05	2.08	0.50	-2.63
Ni	0.22	2.08	0.06	-1.92
Fe	0.56	2.08	0.10	-1.62

Table A2. Calculated heat of solution (H_S , eV), structural (H_{SC} , eV) and chemical (H_{CC} , eV) contributions of Ti, Ni, and Fe in the defective Cu crystal structures.

Solutes	H_S (eV)	H_{SC} (eV)		H_{CC} (eV)
		E_{HR}	E_{SS}	
Ti	0.08	0.90	0.24	-1.06
Ni	0.35	0.90	0.02	-0.57
Fe	0.73	0.90	0.02	-0.20

Comment 8: Convergence tests for DFT parameters (k-point mesh, cutoff energy) are absent, casting doubt on the reliability of calculated heats of solution (H_S).

Response 8: Thank you for your comment. We have conducted related convergence tests (Figure A10) to validate the reliability of the calculated heats of solution. It can be seen that atom energy remains stable when the cut-off energy exceeds 400 eV (K -point: $7 \times 7 \times 7$) and the K-point is beyond $4 \times 4 \times 4$ (cut-off energy: 450 eV).

Related revisions 8:

Main text: In addition, the cut-off energy and k-point were tested for convergence (Figure A10).

(Lines 443-444 in the main text)

**Supplementary material:** Convergence tests were conducted to validate the reliability of
calculated heats of solution (Figure A10a and b). It can be seen that atom energy remains stable
when the cut-off energy exceeds 400 eV (K -point: 7×7×7) and the K-point is beyond 4×4×4 (cut-
off energy: 450 eV). (Lines 185-190 in the SM)

Figure A10. The convergence test of the cut-off energy(a) and k-points (b).

**Comment 9:** Generalizability of C_L and C_R : The claim that these parameters universally predict
wetting behavior should be tempered, as non-ideal systems (e.g., short-range ordered alloys) may
deviate.

**Response 9:** Thank you for your suggestion. We acknowledge that deviations may occur in non-
ideal systems, such as short-range ordered alloys. We then toned down our claims in the revised
paper.

From a thermodynamic perspective, non-ideality reflects that heteroatomic interactions (ϵ_{AB}) differ
strongly from homoatomic ones (ϵ_{AA} , ϵ_{BB}), leading to very positive or negative λ_{AB} values ($\lambda_{AB} =$
$ZN_a \left[\epsilon_{AB} - \frac{\epsilon_{AA} + \epsilon_{BB}}{2} \right]$, Z is the mean coordination number in the liquid, N_a is the Avogadro's
number. ϵ_{AB} , ϵ_{AA} , ϵ_{BB} are negative). According to the Miedema model, large positive λ_{AB}
corresponds to weak A–B bonding and immiscibility (e.g., Cu/W, Cu/Mo), where spreading remains
rapid, while very negative λ_{AB} favors IMC formation (e.g., Al/Ni, Al/Fe) and slow, IMC-controlled
spreading (see **Response 17**).

**Related revision 9:**

**Title:** Thermodynamic framework for assessing dissolutive wetting behaviors in metallic systems
(Lines 1-2 in the main text)

**Results section:** It is found that wetting behaviors in dissolutive systems strongly correlate with the
combination of C_L and C_R . (Lines 94-95 in the main text)

**Discussion section:** In conclusion, we unraveled different wetting behaviors and proposed a
thermodynamic approach for assessing wetting properties in dissolutive metallic systems based on
C_L and C_R . (Lines 362-364 in the main text)

4. Discussion and Conclusions

Mechanistic Evidence Lacking:

**Comment 10:** The proposed step-flow mechanism relies on indirect observations (e.g., sharp edges
in quenched samples). Direct evidence (e.g., in situ composition/velocity measurements at the
contact line) would strengthen the argument.

**Response 10:** Thank you for your suggestions. The proposed step-flow mechanism comprised four
processes: (1) pinning the CL (Figure A11a). (2) precursor film formation ahead of CL (Figure
A11b). (3) Marangoni flow toward CL and liquid accumulation (Figure A11c). (4) overflowing of
the CL (Figure A11d). We have in-situ observed every step of this mechanism and provided direct
evidence in our previous publication (Sun et al. Acta Materialia 282 (2025): 120519; Cheng et al.
Chemical Engineering Journal (2025): 165377)^{11,12}.

**Figure A11.** Illustration of the novel step flow mechanism proposed in this work, where (a)
indicates the droplet CL is pinned by the solid solution layer formed at the interface due to the
dissolution. (b) reveals the solutal Marangoni flow on the surface caused by a surface tension
gradient, and the flow will drive the fluid toward solidified edge. (c) shows the accumulation of
liquids at the edge with circular flows leads to (d) the sudden overflow of droplet CL.

(1) Pinning the CL

The CL is pinned by the solidified edge due to the formation of solid solution layer at the S/L
interface. The solidification of the Cu droplet is investigated by in-situ observation of (in the CSLM).

The droplet CL is pinned 50 s after the start the isothermal stage at 1100 °C (Figure A12a). After
 600 seconds, a wide solidified edge has developed at the CL (Figure A12b). During the rapid cooling
 stage, this solidified edge almost remains unchanged when the liquid phase becomes first partially
 solidified at 1030 °C (Figure A12c) and then completely solidified at 780 °C (Figure A12d). The
 SEM/EDX analysis (Figure A12e and f) reveals that the Ni composition near the interface is 22.6
 at. % (point 1) (2.1 at. % (point 2), away from the interface). The Cu-Ni binary phase diagram
 suggests that the region at both points 1 completely solidified before quenching. In comparison, the
 liquid fractions at point 2 is 100 % based on the level rule of the phase diagram. This indicates that
 the dissolution of Ni near the L/S interface leads to an isothermal solidification and formation of
 Cu-Ni solid solutions. With extended isothermal time, more dissolution of Ni results in the
 development of the solid solution in a direction perpendicular to the L/S interface. The solidified
 edge then pins the movement of the CL.

**Figure A12.** The in-situ observation of Cu on Ni indicates no obvious solidified edge is observed
 (a) at 50 s at 1100 °C, while a wide solidified edge is developed at the CL in (b) at 600 second.
 During rapid cooling stage (see Video S1), the solidified edge almost remains unchanged when
 liquid is partially solidified at 1030 °C (c), or completely solidified at 780 °C (d). The bended region
 is observed from cross section in (e). The EDS analysis (f) indicates the Ni enrichment at the L/S
 interface.

(2) Precursor film formation

A thin precursor film protruding from the solidified edge is observed and 1150 °C (as exemplified
 at 600 s in Figure A13a). And the quenched droplet (Figure A13b) confirms the existence of a
 precursor film (indicated by the white arrows) with a Cu composition of 14 at. % (spot 1). The
 significant presence of Cu in the film as can be derived from the EDS mapping (Figure A13c)

strengthening the hypothesis that the precursor film is an extension of the bulk liquid Cu ahead of
 the solidified edge. SEM pictures in both top view and cross-section (Figure A13d) reveal that the
 width of the precursor film is 30 μm with a ridge found at its end. Figure A13e further confirms the
 presence of Cu (spot 2, 14 at. %) even 30 μm away from the edge. Saiz et al.¹³ speculated that the
 Marangoni flow towards the CL promotes the formation of a precursor film in Au/Ni systems.
 Specifically, the dissolution of Ni into Au at L/S interfaces induces a surface tension gradient,
 driving the liquid from the droplet apex towards CL and resulting in the formation of a Marangoni
 film. They further proposed that a triple line ridge can form at the end of the film and pin the wetting
 liquid front. In our cases, the substantial Ni presence, as evidenced by the isothermal solidification
 at the L/S interface, may also induce a Marangoni flow towards the solidified edge.

 **Figure A13.** The precursor film formed in front of the solidified edge at (a) 1150 °C. The precursor
 film at 1150 °C is quenched and indicated in (c) the SEM picture. The SEM picture in cross section
 in (d) indicates a film with a width of 30 μm and a hump at the end. The EDS analysis within the
 hump in (e) confirms the presence of Cu.

(3) Marangoni flow

We characterized the Marangoni flow towards the solidified edge by observing the movement of
 alumina inclusions segregated at liquid/gas interfaces of Cu melt on Ni at 1100 °C. Due to the
 buoyancy force and poor interaction with liquid metals, alumina particles will segregate on the
 liquid surface to act as trackers of the surface flow¹⁴. Effects of the alumina particles on the liquid
 viscosity and surface tension can be ignored as they do not tend to distribute within and dissolve
 into the droplet¹⁵. In Figure A14a, no alumina particles are observed at the liquid/gas interface of
 the droplet during a short isothermal stage of 50 seconds. However, at 250 seconds, an accumulation

of alumina clusters is found near the CL in Figure A14b. The accumulation is not caused by
 attraction forces between particles, such as capillary force, since the particles travel distances greater
 than the effective range of these forces (e.g. $< 100 \mu\text{m}$)¹⁶. Instead, these clusters are formed as a
 result of the agglomeration of alumina particles associated with a liquid flow towards the solidified
 edge. The lines (blue, green, and red) indicate the path followed by the particles. At this moment,
 the droplet CL remains pinned by the solidified structure, but an ongoing flow towards the edge
 leads to the accumulation of more alumina particles and liquid. As a result of this dynamic stacking
 of liquid at the edge, the entire Cu droplet suddenly overflows the solidified edge at 300 seconds
 (Figure A14c), with alumina clusters moving towards the newly formed CL (Figure A14d). The
 SEM picture after quenching (Figure A14e) reveals the distribution of these alumina clusters along
 the CL. The subsequent EDS mapping of the Al and O elements confirms that these clusters are
 indeed formed by the agglomeration of alumina particles (Figure A14f). Another piece of evidence
 about the surface flow is shown in Figure A14g-i. A solidified particle is found near the edge at 1150
 453 °C, indicated by the red dotted circle in Figure A14g. However, it almost disappears immediately
 after 1 second while the droplet CL remains pinned (Figure A14h). This large particle is expected
 to be a protruding part of solid Cu-Ni solution. It is highly unlikely for the large solid Cu-Ni particles,
 with a length of $40 \mu\text{m}$, to dissolve completely within 1 second, one plausible explanation is that
 the Cu liquid flows towards the solidified edge, submerging this large particle. The droplet CL
 subsequently overflows the edge at 453 second, as depicted in Figure A14i.

**Figure A14.** The solutal Marangoni convection on the surface tracked by Al_2O_3 particles (0.2 wt. %) within the Cu droplets. No Al_2O_3 particles are observed on the surface at 50 s in (a), while an accumulation of Al_2O_3 clusters is found near the contact line at 260 s in (b), with the lines indicating the flow direction (see Video S2). The droplet CL overflows at 300 s in (c) and the Al_2O_3 clusters are pushed toward the newly formed CL in (d). (e) The SEM picture of quenched liquid surface and (f) the EDS mapping confirm the presence of Al_2O_3 particles. (g) The solid particle at the CL is submerged in (h) and the droplet CL immediately overflows in (i). This provides a piece of evidence about the existence of surface flow.

We also employed CFD simulations to further investigate the Marangoni flow and the corresponding element redistribution. Figures A15a indicates the mean velocity contours and streamlines of the droplet in scenarios without and with the Ni-Concentration Induced Marangoni effect (Ni-CIME), respectively. Without the Ni-CIME, the liquid at the bottom center of the droplet first directs towards the CL and then flows along the liquid/gas interface to the droplet apex, leading to an overall counterclockwise direction (indicated by the white arrows in Figure A15a-1). This flow pattern is referred to as natural capillary flow¹⁷. It is also seen that the overall capillary flow is actually composed of multiple vortices in the counterclockwise direction. Compared to Figure A15a-1, with the Ni induced Marangoni effects, the overall flow at the liquid/gas interface (Figure A15a-2) reveals a reverse direction from the droplet apex to the CL. Specifically, the higher Ni concentration at the CL (Figure A15c and d) results in a larger surface tension, while the surface tension at the droplet apex is lower. This surface tension gradient generates a Marangoni shear, leading to a liquid/gas interfacial flow from the droplet apex toward the CL, as confirmed experimentally by tracking the movement of alumina particles. We also found that the formation of sharp edge is related to solutal Marangoni flow, which generates vortices near the solid/liquid interface as shown in Figure A15b. These vortices erode the solidified layer, accelerating the dissolution of Ni into the Cu droplet. Simultaneously, the Marangoni flow transports dissolved Ni toward the CL, forming localized regions of high Ni concentration, as shown in Figure A15c and 15d. This localized enrichment facilitates the formation of a solidification front near the contact line, as shown in Figure A16a. The sharp edge structures form and pin the CL movement, as experimentally observed in Figure A16b and computationally simulated in Figure A16c. Sharp edges are eroded again due to the continuous dissolution accelerated by the Marangoni vortices after overflowing. When the advancing liquid phase overtakes this solidification front, the sharp edge structure is observed (Figure A16d). However, as the wetting process progresses, these structures gradually dissolve due to ongoing interactions with the liquid flow.

**Figure A15.** Mean velocity contours and streamlines in the droplet without (a-1) and with (a-2) Ni-
 concentration induced Marangoni effect (Ni-CIME). (b) The temporal evolution of flow velocity and
 vectors in the droplet. (c) Evolution of Ni concentration along the liquid/gas interface at
 different times. The abrupt increase of Ni concentration after reaching the CL is due to the presence
 of Ni substrates. (d) Contours of Ni concentration within the droplet at different times.

**Figure A16.** Analysis of the formation mechanism of the sharp edge structure during the wetting
 process: (a) Ni concentration contour, (b) sharp edge observed in contact heating experiments, (c)
 simulated sharp edge structure, and (d) the sharp edge structure dissolved after overflowing.

(4) Liquid overflow

The step flow behavior results in the progressive CL advancement. Each step is associated with an
overflow phenomenon and the CL then comes into contact with fresh Ni substrate. This results in
the dissolution of Ni not reaching saturation at the beginning of each step. Consequently, the
Marangoni flow can be continuously fuelled by dissolution, allowing the step flow to persist.

CFD simulations were further performed to clarify the overflow mechanism (Fig. A17a–b). The
wetting process of Cu droplets on Ni at 1150 °C involves three stages—initial pinning, overflow,
and new pinning (Figure A17a). Overflow further divides into spreading and stabilizing sub-phases
(Figure A17b). In the spreading phase, oscillations in base diameter and droplet height reveal
dynamic instabilities: first a vertical stretch (shrinking diameter, rising height), followed by rapid
spreading (expanded diameter, reduced height). In the stabilizing phase, moderate shape fluctuations
were observed as forces, flows, Ni redistribution, and the evolution of solid solution layer gradually
reach equilibrium. The curves for base diameters and droplet height ultimately flatten, indicating a
new pinned stage.

The overflow dynamics are governed by inertia, capillary force, and viscous forces, as reflected by
the dimensionless numbers Re , We and Ca (Figure A17c)¹⁸. Re , We , Ca represent inertia-to-viscous,
inertia-to-capillary, and viscous-to-capillary ratios¹⁹, respectively. In the initial pinning stage, weak
Marangoni convection and the pinning effects yield low Re , We , and Ca . The capillary force then
dominates over other effects as a result of these low numbers. During spreading, Ni redistribution
drives the strong Marangoni flow, increasing Re and pushing $We > 1$. Therefore, inertia temporarily
overcomes capillary force, resulting in droplet vertical stretching (shrinking diameter, rising height).
In the stabilizing phase, Re decreases (though still ≈ 200), We drops below 1, and Ca remains low.
The capillary force controls with limited viscous dissipation. Finally, the system reaches force
balance, yielding a stable contact angle and equilibrium droplet morphology. The droplet then
moves to a new pinned stage.

In summary, we have thoroughly investigated the step flow mechanism and provided direct evidence
through in-situ solidification observations, tracking of Al_2O_3 particles, quenching of the precursor
film and simulating flow behaviors. We believe these findings strongly support the existence of the
mechanism. The step flow mechanism is further confirmed by the comparative study between

equilibrium Cu-10 at. %Ni/Cu-30 at. % Ni and non-equilibrium Cu/Ni systems (see **Response 18**).
 However, in this paper, our focus is broader: examining whether the step flow mechanism
 contributes to altering wetting behaviors. Therefore, detailed explanations are not repeated here, but
 we have referenced our previous publications (Sun et al. Acta Materialia 282 (2025): 120519; Cheng
 et al. Chemical Engineering Journal (2025): 165377) where this evidence is comprehensively
 discussed:

**Figure A17.** (a) The wetting process of Cu on Ni involves initial pinned stage, overflowing and new
 pinned stage, (b) Overflow further divides into spreading and stabilizing sub-phases, (c) the

overflow dynamics reflected by the dimensionless numbers Re , We and Ca .

**Related revisions 10:** More details about step flow behaviors for the Cu/Ni system have been given
in our previous studies^{11,12}. For instance, the formation of a solid solution layer is confirmed by in-
situ observation and quenching¹¹. The effects of dissolution on interfacial energy, solutal Marangoni
flow and interface morphology were characterized experimentally and further investigated by CFD
simulations¹². **(Lines 117-121 in the main text)**

**Comment 11:** The assertion that high C_R (83.9 at. % for Cu/Fe) prevents step-flow is qualitative;
a quantitative threshold or kinetic argument is needed.

**Response 11:** Thank you for your suggestion. We have clarified the quantitative threshold
preventing the step flow mechanism as follows:

**Supplementary material:**

Assuming equilibrium solidification at the liquid/solid metal interface, the fraction of solid solutions
(f_S) in C_R can be estimated using the lever rule

$$554 \quad f_S = \frac{C_0 - C_L}{C_R} = \frac{C_0 - 0.047}{C_R}$$

where C_0 is the Fe composition at the L/S interface. Based on quenched samples, $C_0 \approx 0.05$. As
shown in Figure A18, complete solidification ($f_S \approx 1$) requires $C_R \approx 0.02$. Even for $C_0 = 0.15$, full
solidification still requires $C_R \approx 0.10$. In contrast, the actual value $C_R = 0.839$ is far above this
threshold, meaning that the CL cannot pin and the step-flow mechanism cannot occur.

We note, however, that this analysis is based on ideal equilibrium conditions. In reality, isothermal
solidification may deviate from equilibrium^{20,21}, and partial solidification (e.g., $f_S \approx 0.8$) could still
introduce “pinning” effects. Even so, the large C_R value of 0.839 clearly precludes activation of the
step-flow mechanism. **(Lines 96-110 in the SM)**

**Related revisions 11:**

**Main text:** For the Cu/Fe system (Case 3), despite a low Fe solubility of 4.7 at. %, the high C_R
(89.2 at. %) impedes the complete solidification even after reaching saturation. At the CL, only 0.8 %
solids form (Figure 2d) with a Fe composition of 5.4 at. % (spot 3 in Figure 2b), while complete
solidification requires a threshold C_R of 2 at. % (Figure A18 in the SM). As a result, no step flow
is observed in Case 3. **(Lines 203-208 in the main text)**

**Figure A18.** The relationship between fraction of solid solutions and C_R with different Fe
 compositions at the L/S interface.

**Comment 12:** Weak Link to Applications: While the conclusion mentions implications for
 welding/coating, no concrete examples or guidelines for process optimization are provided.

**Response 12:** We thank the reviewer for this insightful suggestion. We believe that our conclusions
 can apply for process optimization, particularly in dissimilar metal systems. Such systems combine
 the advantages of different materials to meet the growing demands for high-performance and
 multifunctional components. Representative examples include Cu–W²², Cu–Fe²³, Cu–Ti–Fe²⁴, Cu–
 Ti–Ni²⁵, Sn–Bi²⁶, and Sn–Au⁷, which are increasingly used in electrical contacts, thermal
 management, aero-engine components, and microelectronics (Table A3). Their processing methods
 such as high-temperature infiltration, liquid-phase sintering, and cold metal transfer welding, are all
 fundamentally governed by wetting behavior, which directly determines the final properties.

To reflect this point, we have provided one concrete example and revised the conclusion:

Table A3. Field of applications of dissimilar metal systems

Systems	Materials	Processing techniques	Properties	Applications
Cu-W	Cu alloy-W ^{22,27}	Infiltration	High thermal conductivity; Low thermal expansion	Electric contact material Thermal management
Cu-Fe	Cu-316L ²⁸	Liquid-phase	High thermal conductivity;	Aeroengine fuel

	Cu-M300 ²³	sintering Infiltration	High strength	nozzle
Cu-Ti- Fe	CuSi3-TC4- 304 ²⁴ ; CuCrV-TC4- 316L ²⁹	Cold metal transfer; Gas tungsten arc welding	High thermal conductivity; High strength; Low density; Corrosion resistant	Seawater condenser
Cu-Ti- Ni	CuSi3- INX750- TC4 ²⁵	Gas tungsten arc welding	High thermal conductivity; High strength; Low density; Corrosion resistant	Components for aeroengineer combustion chamber
Sn- metals	Sn-Bi ^{6,26,30} Sn-Ag ³¹ Sn-Au ⁷	Soldering	High electric conductivity; High thermal conductivity	microelectronics fabrication

**Related revisions 12:** The proposed mechanism offers guidance for optimizing high-temperature
processes (e.g., welding, coating, infiltration). For example, during the high-temperature infiltration
Cu–diamond composites, diamond metallization is required to improve wettability with Cu³². Ni
coatings may improve bonding but slow infiltration, Fe coatings enable fast infiltration but weak
adhesion, while Ti coatings may balance both. **(Lines 369-373 in the main text)**

5. Formatting and Clarity Issues

Figure/Tables Need Improvement:

**Comment 13:** Table 1's symbols (e.g., " H_s (eV)") should follow consistent formatting (italic/roman).

**Response 13:** Thank you for your suggestion, we have unified the formatting as italic in Table 1 in
our revised paper.

**Related revision 13:**

Table 1: Calculated heat of solution (H_S , eV), structural (H_{SC} , eV) and chemical (H_{CC} , eV)
contributions of Ti, Ni, and Fe in Cu crystal structures. **(Lines 291-292 in the main text)**

Solutes	H_S (eV)	H_{SC} (eV)		H_{CC} (eV)
		E_{HR}	E_{SS}	
Ti	0.03	1.38	0.16	-1.51
Ni	0.28	1.38	0.02	-1.12
Fe	0.65	1.38	0.01	-0.74

**Comment 14:** Figure 1b's schematic labels are too small to read.

**Response 14:** Thank you for your suggestion, we have enlarged schematic labels in Figure 1b in
the revised paper.

**Related revision 14:** **(Lines 133-138 in the main text)**

**Comment 15:** Termconsistencies: "Step flow" is alternately called "step-wise spreading";
 standardize usage.

**Response 15:** Thank you for your suggestion, we have unified as "step flow" in our revised paper.

**Related revision 15:** We observed a slow step flow behavior in systems with a relatively low C_L
 and narrow C_R . For systems with high C_L , the spreading begins relatively fast and transitions to the
 step flow mechanism when C_L is reached. **(Lines 27-30 in the main text)**

Spreading in the step flow mechanism at the beginning of the isothermal stage for Cu/Ni (Case 1).

**(Lines 128-129 in the main text)**

Spreading in the step flow mechanism after a period of time (~2000 s) for Cu/Ti (Case 2). **(Lines**

**130-131 in the main text)**

**Comment 16:** Redundant References: Lines 532–533 cite the same study twice with nearly identical

text.

**Response 16:** Thank you for your comment, we have corrected and double checked our reference
in our revised paper.

**Comment 17:** The authors focus on purely dissolutive wetting, yet the systems they studied are not
purely dissolutive. The selected representative systems involve both dissolution and the saturation
precipitation of intermetallic compounds (IMCs), and the influence of the latter cannot be entirely
ruled out.

**Response 17:** Thank you for your question. We would like to address this comment from two
aspects:

**(1) Classification of dissolutive systems:**

We agree that the term “dissolutive systems” more appropriately describes our systems than “purely
dissolutive systems”, as it is a broader and more inclusive concept. According to the classification
by Singler et al.³³, dissolutive systems can be divided into two categories: (1) purely dissolutive
systems, where dissolution is the sole liquid-substrate interaction. Examples from our study include:
Cu-Ni, Cu-Fe, Au/Pt, Ag/Au (at 1150 °C), and Ag/Cu (at 1000 °C); (2) product forming systems at
temperatures exceeding the stability temperatures of all products, e.g., Cu/Ti Ag/Ni, Ag/Ti (at
1150 °C), Sn/Cu (at 800 °C) and Sn/Fe (at 950 °C).

We argued that IMC formation does not affect our wetting behavior for the following reasons:(1) At
the temperatures investigated, IMCs are thermodynamically unstable, and only solid solutions are
expected; (2) In non-contact heating experiments, the droplet and substrate are physically separated
until the desired temperature is reached, preventing IMC formation during heating; (3) For contact
heating, a rapid heating rate of ~300 °C/min is used, minimizing any possible IMC formation during
the heating-up stage.

The second type of dissolutive behavior (i.e., product forming systems behaving as dissolutive at
high temperatures) has also been observed in the literature. For instance, Sun et al.⁴ reported that
Ti₂Cu forms once the Ti solubility in liquid Cu exceeds 62.2 at. % at 1000 °C. However, at 1030 °C
and 1060 °C, no IMC formation was observed in Cu/Ti systems, and the wetting was governed by
dissolution. Protsenko et al.^{34,35} noted that although several IMCs exist in the Cu–Si system, they
are unstable above 860 °C. Thus, at 1100 °C, only equilibrium between liquid Cu-Si alloy and solid

Si occurs, consistent with dissolutive wetting.

We also provide direct experimental evidence: (a) Cu/Ti systems at 1000 and 1150 °C: No IMC
layer is observed at the liquid/solid interface at 1150 °C compare to that at 1000 °C (Figure A19a,
Ti₂Cu IMC layer, picture is derived from Sun et al.⁴); instead, only dendritic phases formed at
cooling stage exist (Figure A19b). EDS mapping of Cu and Ti (Figure A19c and d) confirms the
absence of stable IMCs at the liquid/solid interface.

**Figure A19.** SEM images of Cu/Ti show (a) a Ti₂Cu IMC layer at the L/S interface at 1000 °C⁴,
while (b) only dendritic phases formed during cooling are present at 1150 °C, confirmed by (c) Cu
and (d) Ti mapping. For Sn/Fe, (e) an FeSn IMC layer forms at 650 °C, whereas (f) at 950 °C only
solid solutions are observed, with deep grooves resulting from mutual dissolution.

(b) Sn/Fe systems at 650 and 950 °C: This system transitions from IMC-forming at 650 °C to
dissolutive at 950 °C. At 650 °C, a FeSn IMC layer forms, resulting in a flat L/S interface (Figure
A19e). At 950 °C, only solid solutions are observed, and deep grooves are formed due to mutual
dissolution (Figure A19f). The IMC layer acts as a barrier for dissolution and thus mediating further
dissolution of the substrate metal.

Based on both thermodynamic considerations and experimental evidence, we can confidently state
that all representative systems selected in this study behave as dissolutive systems at the targeted
experimental temperatures.

(2) Effects of IMC formation on wetting

The lateral growth of IMCs at the L/S interface is considered to determine the wetting behavior of
liquid metals and alloys in product forming systems^{2,36}. A longer spreading stage is expected due to
the required time for liquids to reach equilibrium on solids³⁷. The IMC can form by reaction either
between bulk liquid and solid substrates or between the alloying element within bulk liquid and
substrates. The reaction rate, which in turn determine spreading rates, is controlled by the diffusion
of reacting species in the liquid alloy to, or from, the contact line and by local reaction kinetics at
the contact line. Diffusion-controlled wetting³⁸ and reaction-controlled wetting^{3,39} are then defined
based on these respective determining processes. The slower process of these two eventually
controls the overall reaction dynamics.

The dissolutive wetting and product forming systems have been comparatively studied both in our
work and in the literature:

**(a) From our work:** We explain how S/L interactions affect spreading by taking advantages of
Sn/Fe systems which can change from compound-forming systems at 650 °C to dissolutive
scenarios at 950 °C (Figure A20a). At 650 °C, we observe slow, linear spreading at a rate of 0.15–
0.17 μm/s, with only a single instance of contact line (CL) overflow (Figure A20b). In-situ
observations provide direct evidence that the advancing CL is arrested and governed by the lateral
growth of a solidified edge (Figure A20(c1) - (c4)). Further SEM analysis confirms that the edge
originates from an FeSn IMC layer (Figure A20(e1) - (e4)). Spreading behaviors differ at 950 °C.
Although the CL remains stationary during the isothermal stage, the droplet had already rapidly
spread during the heating stage. In-situ imaging reveals the formation of “white meshes” ahead of
the CL (Figure A20(d1)-(d4)), and SEM analysis of quenched samples identifies these meshes as
liquid Sn spreading along surface grain boundaries (Figure A20(f1)-(f4)).

Overall, for product forming system, the lateral IMC growth controls the spreading process. The
linear dependence of spreading on time conforms to the reaction limited model^{3,40} and the spreading
rate is in the same order with that in reactive Sn/Au systems (less than 10 μm/s)⁷. Compared to the
step flow mechanism, the rapid chemical reactions in Sn/Fe systems leads to a stronger solid-liquid
interaction and consumes most Sn at the CL. Consequently, the CL can hardly overflow by
overcoming the physical/chemical/energy barrier due to the IMC formation. For dissolutive systems,

the spreading is fast due to a large C_R . The liquid Sn can even wet Fe surface along grain boundaries
 ahead of the droplet without an arresting CL. A more detailed discussion of grain boundary wetting
 will be presented in a forthcoming publication.

**Figure A20.** (a) Sn–Fe phase diagram showing the transition from compound-forming to dissolutive
 wetting between 650 and 950 °C. (b) CL advancement during the isothermal stage. At 650 °C (c1–
 c4), lateral growth of a ‘solidified edge’ and droplet overflow are observed, while at 950 °C (d1–
 d4) no significant edge or CL arresting. SEM images of quenched samples confirm that the
 solidified edge at 650 °C (e1–e4) is the FeSn IMC. At 650 °C (f1–f4), liquid Sn even wets Fe grain
 boundaries ahead of the droplet without CL pinning.

**(b) From the literature:** For product forming Cu/Ti systems, spreading is slowed down by the
 formation of a Ti₂Cu IMC layer at 1000 °C⁴. At higher temperatures (1060 and 1150 °C, data for
 1060 °C are from Sun et al.⁴), the spreading for dissolutive Cu/Ti systems is rapid, and conforms to
 systems with the late presence of step flow mechanism (Figure A21a). Similarly, comparison of

dissolutive Sn/Bi and product forming Sn/Au systems confirms that spreading is fast in Sn/Bi but
much slower in Sn/Au due to the lateral growth of an IMC ‘foot’ or ‘film’(Figure A21b)⁵.

**Figure A21.** (a) Cu/Ti systems show rapid spreading in dissolutive regimes (1060⁴ and 1150 °C)
but much slower spreading in the product forming regime (1000 °C⁴). (b) Comparison of dissolutive
Sn/Bi and product forming Sn/Au systems: spreading is fast for Sn/Bi, but slower for Sn/Au due to
lateral IMC “foot” or “film” growth⁵.

**Related revisions 17:** Compared with product forming systems with intermetallic compounds
(IMCs) forming at the L/S interface, dissolution is the primary interfacial activity in dissolutive
systems. Dissolutive wetting behavior can be further categorized into two main types³³: (1) purely
dissolutive systems, where dissolution is the sole liquid-substrate interaction (e.g. Sn/Bi²⁶ and
Cu/Ni¹¹) (2) product forming systems at temperatures exceeding the stability limits of all products
(e.g. Cu/Ti at 1060°C⁴ and Cu/Si at 1100°C³⁴). **(Lines 44-50 in the main text)**

With the late presence and absence of the step flow mechanism, the Cu droplets can rapidly spread
on Ti and Fe substrates due to the strong metal-metal atom affinity. This is similar to the product
forming model, where the lateral growth of IMCs at L/S interface arrests the CL and results in a
slow spreading¹⁻³. For instance, Sun et al.⁴ reported that the lateral growth of Ti₂Cu leads to a slow
spreading for the Cu/Ti system at 1000 °C. Similarly, A comparison of dissolutive Sn/Bi and product
forming Sn/Au systems shows fast spreading in Sn/Bi (conform to Case 2) but slower kinetics in
Sn/Au at 250 °C due to IMC ‘foot’ growth^{6,26}. However, spreading in Sn/Au accelerates at 450 °C,
where higher Au solubility (81 at. %) delays IMC formation⁷. Thus, early CL arrest from step flow
or IMC growth contributes to slow spreading. In contrast to the "pinning–overflow" mode of the
step flow mechanism, spreading kinetics in product forming systems is typically linear¹⁻³. **(Lines**
**307-318 in the main text)**

**Comment 18:** For purely dissolutive wetting, the establishment of a thermodynamic framework

should generally be based on the following two aspects:

First, the changes in solid/liquid interface geometry caused by dissolution must be considered.

Second, the effect of dissolved solutes (whether in the liquid phase or as solid solutions) on
interfacial tension must be accounted for.

These key points were not adequately addressed by the authors. To properly study purely dissolutive
wetting, the authors should focus on the following

[1] L. Yin, Reactive wetting and spreading in binary metallic systems, Philosophy in Materials
Engineering in the Graduate School, Binghamton University, New York, 2005, p. 188.

[2] L. Yin, A. Chauhan, T.J. Singler, Reactive wetting in metal/metal systems: Dissolutive versus
compound-forming systems, Materials Science and Engineering: A 495(1-2) (2008) 80-89.

[3] L. Yin, et al., Reactive wetting in metal–metal systems, Journal of Physics: Condensed Matter
21(46) (2009) 464130.

[4] L. Yin, S.J. Meschter, T.J. Singler, Wetting in the Au-Sn System, Acta Materialia 52(10) (2004)
2873-2888.

[5] L. Yin, B.T. Murray, T.J. Singler, Dissolutive wetting in the Bi-Sn system, Acta Materialia 54(13)
(2006) 3561-3574.

[7] J.A. Warren, W.J. Boettinger, A.R. Roosen, Modeling reactive wetting, Acta Materialia 46(9)
(1998) 3247-3264.

[8] R. Warren, Solid-liquid interfacial energies in binary and pseudo-binary systems, Journal of
Materials Science 15(10) (1980) 2489-2496.

**Response 18:** Thank you for your suggestion and the valuable references. We agree that dissolution
at L/S interfaces can alter both interfacial morphology and interfacial energies. From the references
you provided, we can elaborate on several points aligning well with your own conclusions:

**(1) Pinning effects:**

In dissolutive systems with large solubility (e.g., Sn/Bi²⁶, Sn/Au⁷ at 400 °C), spreading is rapid and
follows a trend similar to the universal correlation for inert wetting systems. By contrast, in systems
with low solubility where solid IMCs easily form at the interface (Sn/Au at 250 °C), the CL can be
arrested, resulting in much slower spreading (Table A4).

**Table A4.** Comparative study between the Bi-Sn and Au-Sn systems⁵

Characteristics	Sn/Bi (Purely-dissolutive)	Sn/Au (Compound forming)
-----------------	----------------------------	--------------------------

Interface geometry	Smooth CL, Curved L/S interface.	Roughening CL, Relatively flat interface.
Spreading rate	Rapid spreading (2 cm/s)	Slow spreading (10^{-2} cm/s)
Dissolution behavior	Dissolution well tracks the CL, no diffusion barrier.	IMC impedes dissolution; spreading occurs in the form of “foot”.
IMC effects	None	Pins CL; acts as diffusion barrier; enables “foot” spreading.

Even within the same Sn–Au system⁷, the solubility and IMC formation at different temperatures
play a critical role. At lower temperatures, the solubility of Au in liquid Sn is limited (~10 at. %),
and Au–Sn IMCs form easily at the liquid/solid interface. Consequently, The CL is rapidly arrested
by a “roughening foot” and spreading slows down significantly, with a maximum rate of $\sim 10^{-2}$ cm/s.
The IMC layer impedes further dissolution as a barrier layer³⁷. At higher temperatures, however, Au
solubility in Sn increases (~81 at. %), and IMCs form only after the solubility limit is reached. As a
result, the CL is not arrested in the early stage, and the maximum spreading rate can reach ~ 2 cm/s
(Table A5). In our study, we also observed a slow stepwise spreading in systems with a relatively
low C_L and narrow C_R as a result of rapid isothermal solidification at the CL. The isothermal
solidification is difficult for systems with large C_L or C_R , leading to a rapid spreading.

**Table A5.** Comparison of Sn/Au systems at different temperatures⁷

Characteristics	Low temperature (250-280°C)	Medium temperature (315-415°C)	High temperature (430°C)
Solubility	Low solubility (10 at. % Au)	Medium solubility (30-45 at. % Au)	High solubility (81 at. % Au)
Spreading rate	Slow: $< 10^{-2}$ cm/s	High: 10 cm/s	High: 2 cm/s
IMC evolution	γ -phase lateral growth, pinning contact line	δ -phase transient formation/dissolution (<350 ms)	ζ -phase isothermal formation after reaching solubility
Interface morphology	Foot structure	Interface distortion, thin-layer ahead of CL	Central dissolution well + thick film (20 μ m)

To further validate the “pinning-overflow” characteristics of the step flow mechanism, we
comparatively examined the spreading behavior of equilibrium Cu-10 at. % Ni/Cu-30 at. % Ni and
non-equilibrium Cu–Ni systems. Experiments were conducted with pre-saturated Cu–Ni droplets

on pre-saturated Ni–Cu substrates at 1150 °C. The travelling distance of the CL revealed that the
 step flow mechanism was almost eliminated in the equilibrium system compared to the non-
 equilibrium system (Figure A22a), as confirmed by the in-situ observation during isothermal stage
 (Figure A22b). The quenched droplet at 3600 s is flattened, indicating that spreading had largely
 occurred during heating (Figure A22c).
 EDS analysis near the edge showed a Ni concentration of 22.7 at. % near the CL in the equilibrium
 system (Figure A22c and g). Moreover, EDS mapping demonstrated a homogeneous distribution of
 Ni and Cu between droplet and substrate, and the sharp edge evident in non-equilibrium Cu/Ni
 systems (Figure A22d and e) was much less pronounced in equilibrium system (Figure A22c and f).
 These results suggest that equilibrium conditions suppress dissolution at the liquid/solid interface,
 thus eliminating Marangoni flow and isothermal solidification, and ultimately diminishing the
 occurrence of the step flow mechanism.

**Figure A22.** Comparative study of equilibrium Cu–10 at. % Ni / Cu–30 at. % Ni and non-
 equilibrium Cu/Ni systems. (a) Travelling distance of the CL indicates that step flow behavior is
 nearly eliminated in equilibrium systems, as confirmed by (b) the in-situ observation. SEM reveals

a less evident sharp edge and more uniform Cu–Ni distribution in (c) equilibrium than in (d) non-
 equilibrium systems. The presence of the sharp edge is further confirmed in side views of (e) non-
 equilibrium and (f) equilibrium samples. (g) EDS point analyses.

We further performed wetting experiments to compare spreading behavior in equilibrium and non-
 equilibrium systems. In the equilibrium Cu–10 at. % Ni/Cu–30 at. % Ni system, the spreading is
 rapid (Figure A23a), and SEM images after cooling revealed a flat L/S interface (Figure A23b). EDS
 point analysis showed a Ni concentration of 29.2 at. % near the interface, and EDS mapping
 confirmed a relatively uniform Ni distribution between droplet and substrate (Figure A23c). In
 contrast, the spreading is much slower for non-equilibrium Cu/Ni system due to the introduction of
 the step flow mechanism (Figure A23a). Here, the L/S interface is curved (Figure A23d), with the
 Ni concentration near the interface increasing to 38.8 at. %, and a pronounced composition gradient
 was observed from substrate to droplet (Figure A23e).

**Figure A23.** (a) Non-contact heating experiments show faster spreading in equilibrium Cu–10 at. %
 Ni/Cu–30 at. % Ni systems than in non-equilibrium Cu/Ni systems. (b) SEM after cooling reveals
 a flat L/S interface and (c) EDS shows a more homogeneous Ni distribution. In contrast, (d) the
 non-equilibrium system exhibits a curved L/S interface and (e) a more heterogeneous Ni distribution.

Based on EDS line analysis: in the non- equilibrium system, the Ni concentration decreases from
44 at. % at the interface to 25 at. % at 180 μm away, whereas in the equilibrium system it decreases
only from 30 at. % to 18 at. % over the same distance (Figure A24).

**Figure A24.** The EDS line analysis indicates an insignificant composition gradient compared with
the non-equilibrium system.

We deduce that the equilibrium state suppresses Ni dissolution at the L/S interface, resulting in an
insignificant composition gradient compared with the non-equilibrium system. Consequently, Ni-
induced Marangoni flow, isothermal solidification and the “pinning-overflow” mechanism are
largely eliminated, leading to a rapid spreading.

In conclusion, from both literature and our comparative study of equilibrium and non-equilibrium
Cu–Ni systems, we can confirm that the ‘pinning-overflow’ mode induced by the step flow
mechanism, is the primary origin of slow spreading. The presence of pre-existing Ni in droplets or
the usage of pre-saturated Cu–Ni substrates significantly influences the introduction of this
mechanism. These findings suggest that spreading dynamics can potentially be controlled by tuning
the occurrence of the step flow mechanism. This perspective warrants further exploration in a
forthcoming study.

**(2) Interface energies and capillary flow effects:**

The dissolution of substrate elements into liquids may alter interface energies and influence the

capillary flow, thus enhancing or reducing spreading depending on whether the dissolved elements
are tensio-active³⁷. Meanwhile, the capillary flow will influence the distribution of substrate
elements within the liquid droplet. The dissolved elements can change interface energies by the
chemical contribution due to excess free energy and structural contribution caused by structural
mismatch at the interface⁴¹. We also considered the chemical and structural contributions in our
atomistic simulations. On the other hand, the altering interface energy may induce solute-capillary
flow, which in turn redistribute solute within droplets. For example, Cu dissolution in Ag–Cu
droplets accelerates spreading⁴², whereas Si dissolution in Cu induces a reverse flow from the CL
to the droplet apex, suppressing wetting³⁵. The formation of precursor films or solute redistributions
due to Marangoni flow have been reported in Au/Ni¹³ and Sn–Ta–V⁴³ systems. Yin et al.²⁶ asserted
that the global mass transport in the drop was determined to be dominated by convective transport
at early and intermediate times, with the consequence that dissolved solute was confined to
horizontal layers along the S/L interface; diffusive transport dominated at late times.

We have already considered the effects of dissolution on interface energy and capillary flow
behavior in our previous published paper (Sun et al. *Acta Materialia* 282 (2025): 120519; Cheng et al.
*Chemical Engineering Journal* (2025): 165377)^{11,12}: On the one hand, we have experimentally
confirmed the presence of a solutal Marangoni flow by tracking the movement of alumina particles
at liquid/gas interfaces (see **Response 10**). We also employed CFD simulations to investigate the
flow mechanism within the droplet and at the L/G interface.

Figures A25a indicates the mean velocity contours and streamlines of the droplet in scenarios
without and with the Ni-Concentration Induced Marangoni effect (Ni-CIME), respectively. Without
the Ni-CIME, the liquid at the bottom center of the droplet first directs towards the CL and then
flows along the liquid/gas interface to the droplet apex, leading to an overall counterclockwise
direction (indicated by the white arrows in Figure A25a-1). This flow pattern is referred to as natural
capillary flow¹⁷. It is also seen that the overall capillary flow is actually composed of multiple
vortices in the counterclockwise direction. Compared to Figure A25a-1, with the Ni induced
Marangoni effects, the overall flow at the liquid/gas interface (Figure A25a-2) reveals a reverse
direction from the droplet apex to the CL. Specifically, the higher Ni concentration at the CL (Figure
A25c and d) results in a larger surface tension, while the surface tension at the droplet apex is lower.

This surface tension gradient generates a Marangoni shear, leading to a liquid/gas interfacial flow
from the droplet apex toward the CL, as confirmed experimentally by tracking the movement of
alumina particles. Due to the Ni-CIME, it can be seen that the direction of individual vortices also
adopts a reversed direction in clockwise-dominated compared to Figure A25a-1. Moreover, the
magnitude of the flow velocity decreases with the introduction of the Ni-CIME. This is due to the
fact that the overall internal flow induced by the Ni-CIME is counterbalanced by the original
capillary flow in the reverse direction. Figure A25b indicates the temporal evolution of the flow
velocity and vectors in the droplet with the Ni-CIME. The overall internal flow pattern is
characterized by two dominant motions: vortical flow for individual vortices and the downward
movement of vortices along the liquid/gas interface. The direction of the vortex closest to the center
of the droplet preliminarily exhibits clockwise (e.g., 0.16 - 0.17 s) and then transfers to
counterclockwise at 0.17 s and 0.18 s (see the right-side of Figure A25b). This is caused by the fact
that the more homogenous Ni distribution near the axis of symmetry at 0.17 s and 0.18 s cannot
induce the Ni-CIME. High swirling velocities of clockwise rotation are observed over time (see the
left-side of Figure A25b). Besides, vortices move downwards due to a net force towards the CL
under the effects of the surface tension gradient. Both the individual clockwise flow and the
downward movement of vortices facilitate the droplet overflow beyond the pinned CL, resulting in
the step-like advancement of the droplet as observed experimentally. Figure A25c and d show the
evolution of Ni distribution at liquid/gas interfaces and within the droplet, respectively. It can be
seen that Ni concentration near the CL is larger than that far away from the CL, thus inducing a
Marangoni flow towards the CL. The Ni distribution is also intricately related to the internal flow
(Figure A25d). Ni within the droplet is initially entrained by vortices and then driven towards the
CL due to the dominant downward movement of the vortices along the liquid/gas interface. This
implies that the liquid/gas interfacial flow can also lead to a relatively high Ni concentration near
the CL beyond the original effects of Ni dissolution. This heterogeneous Ni distribution, in turn,
intensifies the Marangoni flow and induces the subsequent overflow. Figure A25e shows the
evolution of the solid solution layer at the liquid/solid interface over time. The boundary of the
solidification layer extends towards the CL over time, pinning the droplet movement.

**Figure A25.** Mean velocity contours and streamlines in the droplet without (a-1) and with (a-2) Ni-
 concentration induced Marangoni effect (Ni-CIME). (b) The temporal evolution of flow velocity and
 vectors in the droplet. (c) Evolution of Ni concentration along the L/G interface at different
 886 times. The abrupt increase of Ni concentration after reaching the CL is due to the presence of Ni
 substrates. (d) Contours of Ni concentration within the droplet at different times. (e) Evolution of
 solidification fractions over time. (b), (c), (d) and (e) incorporate the Ni-CIME.

**(3) Interface morphology:**

We agree that interfacial morphology (e.g., dissolution wells, IMC formation) can affect the
 advancement of the CL. However, this influence occurs only when the formation of a dissolution
 well or IMC is sufficient to arrest the CL. In other words, the outcome depends on the competition
 between the CL advancement rate and mass transport processes (e.g., dissolution kinetics, reaction
 rates, IMC formation rate). In systems with rapid isothermal solidification at the CL, the CL can be
 easily arrested, and spreading after the initial stage is dominated by this process. By contrast, in
 systems with slow isothermal solidification, spreading is primarily governed by capillary and
 viscous forces, leading to rapid behavior similar to inert systems³⁶. This may explain why CL
 kinetics in some dissolutive systems can still be described by models originally developed for inert
 wetting (e.g., molecular-kinetic theory^{44,45}, hydrodynamic models⁴⁶). It should also be emphasized
 that even when IMCs or solid solutions do not physically pin the CL, interfacial dissolution and

reactions can still influence spreading by introducing excess Gibbs free energy and changing
interfacial energy (as discussed above). In our paper, the rapid isothermal solidification in Case 1
leads to a slow spreading. The absence or late presence of isothermal solidification results in a much
faster spreading. While slow spreading is also observed in product forming systems, it
fundamentally differs: the contact line is fully arrested by lateral IMC growth, leading to linear and
inherently slower spreading than the “pinning–overflow” characteristics of the step flow mechanism
(See **Response 17**).

In conclusion, the proposed step flow mechanism inherently accounts for the effects of dissolution
on interfacial energy, capillary flow, and interface morphology, as demonstrated through both
experiments and simulations. More detailed analyses of these aspects can be found in our previous
publications (Sun et al. *Acta Materialia* 282 (2025): 120519; Cheng et al. *Chemical Engineering*
*Journal* (2025): 165377). In this revised paper, we have supplemented essential discussions on the
step flow mechanism; however, the primary focus remains on its occurrence across different systems
and the consequent spreading behavior.

**Related revisions 18:** More details about step flow behaviors for the Cu/Ni system have been given
in our previous studies^{11,12}. For instance, the formation of a solid solution layer is confirmed by in-
situ observation and quenching¹¹. The effects of dissolution on interfacial energy, solutal Marangoni
flow and interface morphology were characterized experimentally and further investigated by CFD
simulations¹². (**Lines 117-121 in the main text**)

**Reviewer #2 (Remarks to the Author):**

**Comment 19:** The authors present a comprehensive study of dissolutive wetting in metallic systems.
Two thermodynamic factors have been identified to control the spreading kinetics. The variety of
experimental and modeling techniques have been used to deliver the complete picture of the process,
and enough details have been provided to reproduce the research. The manuscript is well written
and only minor issues have been found such as some digits introduced in the abstract and sentence
grammar on page 10 line 189 should be fixed. Overall, the work represent significant advancement
to the field, and can be accepted after minor revision.

**Response 19:** Thank you for your time and effort in reviewing our manuscript. We also appreciate
your positive feedback on our work, which is very encouraging for us as we continue our research

on the challenging topic of high-temperature wetting. We have carefully reviewed the paper and
have corrected the errors you listed.

**Error:** Some digits introduced in the abstract: In this study, we investigated 8 metallic systems
encompassing a vast range of thermodynamic behaviors. Our findings reveal 3 distinct wetting
behaviors and identify 2 key thermodynamic parameters that control solid solution formation and
then purely-dissolutive wetting behavior.

**Related revision 19:** In this study, we investigated nine metallic systems encompassing a vast range
of thermodynamic behaviors. Our findings reveal three distinct wetting behaviors and identify two
key thermodynamic parameters that control solid solution formation and then dissolutive wetting
behavior. (Lines 22-26 in the main text)

**Error:** Grammar on page 10 line 189 should be fixed: A small C_R value indicates that the liquid
can rapidly and completely solidifies when the dissolved substrate element exceeds C_L .

**Related revision 19:** A small C_R value indicates that the liquid can solidify rapidly and completely
when the dissolved substrate element exceeds C_L . (Lines 188-189 in the main text)

**Reviewer #3 (Remarks to the Author):**

**Comment 20:** The authors combine experiments, thermodynamic calculations, and density
functional theory based atomistic simulations to assess thermodynamic driving forces in dissolutive
wetting systems. They convincingly show that the interplay between wetting, spreading, and inter-
metallic compound formation can be predicted based on two quantities: the substrate solubility in
the droplet liquid (C_L) and the composition range for solidification (C_R). The paper is well written,
and the figures and tables support well the conclusions drawn. I particularly enjoyed the detailed
view provided by many of the figures in both the manuscript and the supplementary material. I
provide mostly minor suggestions below and one suggestion that is less trivial. I save the less trivial
one for last, so it is not presented in its order of appearance in the manuscript.

**Response 20:** We sincerely thank you for your time and effort in reviewing our manuscript, as well
as your encouraging feedback on our work. We have carefully revised the paper in accordance with
your valuable suggestions.

**Comment 21:** The flat curve for Cu/Fe in Fig. 1 is initially confusing, because fast spreading is
expected; perhaps the observation that spreading is already done by the start of the isothermal stage

should be made a bit earlier in the text (e.g., when the figure is first introduced). In general, at that
point, it would be useful to briefly describe any spreading behavior observed during heat up.

**Response 21:** Thank you for your suggestion. We agree that introducing the flat Cu/Fe curve earlier
improves clarity, and we have revised the text and figure captions accordingly.

**Related revision 21:**

**Main text:** Figure 1a indicates the travelling distance of the CL during the isothermal stage (contact
heating) for Cu/Ni ($C_L = 12.0$ at. % and $C_R = 5.2$ at. %), Cu/Ti ($C_L = 71.8$ at. % and $C_R = 16.8$
at. %), and Cu/Fe ($C_L = 4.7$ at. % and $C_R = 83.9$ at. %) systems at 1150 °C. It should be noted that
Cu droplets rapidly complete spreading on Fe substrates upon melting (beyond the temporal
resolution of CSLM), so no CL advancement was observed during the isothermal stage. **(Lines 105-**
**110 in the main text)**

**Picture captions:** Fig. 1: (a) Travelling distances of the contact line during the isothermal stage
(contact heating methods) for the Cu/Ni, Cu/Ti and Cu/Fe systems at 1150°C. The flat curve for
Cu/Fe arises because spreading was already complete upon reaching 1150 °C. (b) Illustration of the
step flow mechanisms for (c) Cu/Ni and (d) Cu/Ti while the (e) Cu/Fe system did not show evidence
of a step flow. **(Lines 134-138 in the main text)**

**Comment 22:** In the Fig. 1 caption "evidences" should be "evidence".

**Response 22:** Thank you for pointing this out, the error has been corrected.

**Related revision 22:** (e) Cu/Fe system did not show evidence of a step flow. **(Lines 137-138 in the**
**main text)**

**Comment 23:** Lines 185-186: The sentence, " C_R accounting for an interval with coexisting liquid
and solid phases [38].", is not clear.

**Response 23:** Thank you for pointing this out, we have rephased this sentence.

**Related revision 23:** C_R represents the compositional range characterized by the coexistence of
liquid and solid phases⁴⁷. **(Lines 184-185 in the main text)**

**Comment 24:** Line 189: "solidifies" should be "solidify".

**Response 24:** Thank you for pointing this out, the error has been corrected.

**Related revision 24:** A small C_R value indicates that the liquid can solidify rapidly and completely
when the dissolved substrate element exceeds C_L . **(Lines 188-189 in the main text)**

**Comment 25:** Line 256: put “contribution” after “chemical”.

**Response 25:** Thank you for pointing this out, the error has been corrected.

**Related revision 25:** This large difference is primarily due to the chemical contribution (H_{CC}) but
not the structural contribution (H_{SC}) according to Table 1. **(Lines 263-265 in the main text)**

**Comment 26:** Line 257: The labels H_{SC} and H_{CC} are reversed.

**Response 26:** Thank you for pointing this out, the error has been corrected.

**Related revision 26:** This large difference is primarily due to the chemical contribution (H_{CC}) but
not the structural contribution (H_{SC}) according to Table 1. **(Lines 263-265 in the main text)**

**Comment 27:** Line 274: “session” should be “section”.

**Response 27:** Thank you for pointing this out, the error has been corrected.

**Related revision 27:** DFT can be utilized to better understand the wetting behaviors of different
metallic systems, as discussed in the following section. **(Lines 280-281 in the main text)**

**Comment 28:** Lines 385-386: the sentence, “The cooling rate during exceeds 500 °C/min [50].”,
needs fixing.

**Response 28:** Thank you for pointing this out, the error has been corrected.

**Related revision 28:** The cooling rate exceeds 500 °C/min. **(Lines 404-405 in the main text)**

**Comment 29:** In Fig. 6 (and in the supplementary material) the authors present data for other
chemical systems (i.e., in addition to Cu/Ni, Cu/Ti, and Cu/Fe). Those data help bolster the
arguments made in the manuscript using the 3 primary test cases (Cu/Ni, Cu/Ti, and Cu/Fe). For
each of the the latter 2 of those cases, data for at least 2 other similar chemical systems (as far as C_L
and C_R) are presented in Fig. 6. For the Cu/Ni case, no additional supporting chemical systems are
explored. To be clear: the authors have done a lot of very elegant work here; nonetheless, it would
be compelling if one more case of low C_R and low C_R were shown to exhibit behavior similar to
Cu/Ni. The authors compare this system to what I think of as reaction limited reactive wetting
systems, and many such systems have been presented in prior literature. So perhaps the authors don't
feel it is necessary to show an additional system. But perhaps they could at least talk about existing
literature for comparable systems and show that analogous behavior is exhibited?

**Response 29:** Thank you for this valuable comment. We agree that extensive literature has shown
that slower spreading can occur when the lateral growth of IMCs at the CL arrests and governs its

movement. However, as our focus in this paper is on dissolutive systems, we have not attempted to
include product forming systems (see **Response 17**). In dissolutive systems, the CL is pinned by
solid-solution formation rather than IMC growth. We do acknowledge, however, that spreading is
also slow in reaction-limited models. To address the reviewer's concerns, we have revised the
manuscript and supplemented our discussion from three perspectives: **(1) Au/Pt dissolutive system:**
We added new experimental results on dissolutive Au/Pt systems (low C_L and low C_R), which show
similar behavior to Cu/Ni. The step-flow mechanism and subsequent slow spreading were also
observed in Au/Pt, directly confirming our thermodynamic framework (Fig. 6 in the revised paper).
**(2) Pre-saturated Cu–Ni system:** We compared the wetting behavior of pre-saturated Cu–Ni
droplets on pre-saturated Ni–Cu substrates with that of non-equilibrium Cu–Ni at 1150 °C. The
former exhibits much faster spreading because the step-flow mechanism is suppressed due to the
absence of dissolution at the L/S interface. **(3) Comparison with compound-forming systems:** We
compared dissolutive and compound-forming wetting in the same system by varying temperature
(e.g., 650–950 °C in Sn/Fe, 1000–1150 °C in Cu/Ti from the literature⁴). Spreading significantly
slowed down when transitioning from dissolutive to compound-forming systems. In the former, fast
spreading is a result of absent or latterly present step flow mechanism. In the latter, rapid IMC
formation at the CL arrests its movement (roughening the CL) and leads to a slow spreading.
Detailed discussions of these three aspects have been discussed below and added to the revised
manuscript.

**(1) Au/Pt dissolutive system**

We also explored the additional Au/Pt system to demonstrate reproducibility particularly in the low
C_L and low C_R regime populated only Cu/Ni in the first version of this paper. Figure A26a shows
the CL advancement during the isothermal stage (contact heating) for an Au/Pt system ($C_L = 4.7$
at. % and $C_R = 7.1$ at. %) at 1150 °C. The step-flow mechanism is observed from the beginning of
the isothermal stage, with one representative step occurring between 1723–1789 s. During this
interval, the CL remains pinned at the solidified edge from 1723 to 1788 s, before being released at
1789 s as liquid Au overflows the edge due to Marangoni flow and liquid accumulation at the CL
(Figure A26b). The Au/Pt system was quenched at 3600 s. The flattened droplet after quenching
indicates good wetting between liquid Au and solid Pt (Figure A26c), with a sharp edge clearly

observed at the CL (Figure A26d). EDS mapping reveals that the solidified edge is Pt-rich (Figure
A26e). EDS Point analysis further shows a Pt concentration of 25.2 at. % at the edge, which
decreases to 8.6 at. % away from the edge (Figure A26d and g). In addition to the top view, the sharp
edge is also confirmed in the side view (Figure A26f). From a thermodynamic perspective, the
quenched Pt composition at spot 4 confirms that the sharp edge results from isothermal solidification,
while regions of the droplet far from the solidified edge remain liquid during the isothermal stage
(Figure A26h).

In summary, the Au/Pt system represents a case similar to the Cu/Ni system, both characterized by
low C_R and low C_L . The spreading mechanism is identical in both systems: C_R controls the onset
of the step-flow mechanism, while C_R determines its timing.

**Related revisions 29:**

**Main text:**

These 9 systems (Cu/Ti, Cu/Ni, Cu/Fe, Au/Pt, Ag/Ni and Ag/Ti at 1150 °C, Sn/Cu at 800 °C, Ag/Cu
at 1000 °C and Sn/Fe at 950 °C) can be categorized into three cases according to the combination
of two thermodynamic parameters: the solubility of the substrate metal within the liquid metal (C_L)
and the solidification composition range (C_R) between the liquidus and solidus line at a given
temperature. **(Lines 72-76 in the main text)**

The proposed mechanism of dissolutive wetting is further confirmed by experimental results from
other systems. For instance, the rapid introduction of step flow mechanism in Au/Pt system aligns
well with Case 1 (Supplementary Figure A26 and Video S3). **(lines 356-359 in the main text)**

**Supplementary material: (Lines 137-147 in the SM)**

**Figure A26.** (a) Travelling distance of the CL vs time illustrating the step flow mechanism of Au

on Pt from the beginning of the isothermal stage at 1150 °C, as confirmed by in-situ observation in

(b). The SEM picture of quenched sample shows (c) a flattened Au droplet and (d) the existence of

the sharp edge at the contact line. The EDS mapping indicates (d) Pt distribution within Au droplets.

The sharp edge is further confirmed by the (f) side view image. The Pt compositions at the positions

near the contact line and at the sharp edge are shown in (g). Corresponding thermodynamical model

in (h) indicates that the sharp edge results from isothermal solidification, while regions of the droplet

far from the solidified edge remain liquid during the isothermal stage.

**(2) Pre-saturated Cu–Ni system**

To further validate the “pinning-overflow” characteristics of the step flow mechanism, we
comparatively examined the spreading behavior of equilibrium Cu-10 at. % Ni/Cu-30 at. % Ni and
non-equilibrium Cu–Ni systems. Experiments were conducted with pre-saturated Cu–Ni droplets
on pre-saturated Ni–Cu substrates at 1150 °C. The travelling distance of the CL revealed that the
step flow mechanism was almost eliminated in the equilibrium system compared to the non-
equilibrium system (Figure A27a), as confirmed by the in-situ observation during isothermal stage
(Figure A27b). The quenched droplet at 3600 s is flattened, indicating that spreading had largely
occurred during heating (Figure A27c).

EDS analysis near the edge showed a Ni concentration of 22.7 at. % near the CL in the equilibrium
system (Figure A27c and g). Moreover, EDS mapping demonstrated a homogeneous distribution of
Ni and Cu between droplet and substrate, and the sharp edge evident in non-equilibrium Cu/Ni
systems (Figure A27d and e) was much less pronounced in equilibrium system (Figure A27c and f).
These results suggest that equilibrium conditions suppress dissolution at the liquid/solid interface,
thus eliminating Marangoni flow and isothermal solidification, and ultimately diminishing the
occurrence of the step flow mechanism.

**Figure A27.** Comparative study of equilibrium Cu–10 at. % Ni / Cu–30 at. % Ni and non-
 equilibrium Cu/Ni systems. (a) Travelling distance of the CL indicates that step flow behavior is
 nearly eliminated in equilibrium systems, as confirmed by (b) the in-situ observation. SEM reveals
 a less evident sharp edge and more uniform Cu–Ni distribution in (c) equilibrium than in (d)
 non-equilibrium systems. The presence of the sharp edge is further confirmed in side views of (e)
 non-equilibrium and (f) equilibrium samples. (g) EDS point analyses.

We further performed wetting experiments to compare spreading behavior in equilibrium and non-
 equilibrium systems. In the equilibrium Cu–10 at. % Ni/Cu–30 at. % Ni system, the spreading is
 rapid (Figure A28a), and SEM images after cooling revealed a flat L/S interface (Figure A28b). EDS
 point analysis showed a Ni concentration of 29.2 at. % near the interface, and EDS mapping
 confirmed a relatively uniform Ni distribution between droplet and substrate (Figure A28c). In

contrast, the spreading is much slower for non-equilibrium Cu/Ni system due to the introduction of
 the step flow mechanism (Figure A25a). Here, the L/S interface is curved (Figure A28d), with the
 Ni concentration near the interface increasing to 38.8 at. %, and a pronounced composition gradient
 was observed from substrate to droplet (Figure A28e). Based on EDS line analysis: in the non-
 equilibrium system, the Ni concentration decreases from 44 at. % at the interface to 25 at. % at 180
 1109 μm away, whereas in the equilibrium system it decreases only from 30 at. % to 18 at. % over the
 1110 same distance (Figure A29).

 **Figure A28.** (a) Non-contact heating experiments show faster spreading in equilibrium Cu–10 at. %
 Ni / Cu–30 at. % Ni systems than in non-equilibrium Cu/Ni systems. (b) SEM after cooling reveals

a flat L/S interface and (c) EDS shows a more homogeneous Ni distribution. In contrast, (d) the
 non-equilibrium system exhibits a curved L/S interface and (e) a more heterogeneous Ni distribution.
 We deduce that the equilibrium state suppresses Ni dissolution at the L/S interface, resulting in an
 insignificant composition gradient compared with the non-equilibrium system. Consequently, Ni-
 induced Marangoni flow, isothermal solidification and the “pinning-overflow” mechanism are
 largely eliminated, leading to a rapid spreading.

**Figure A29.** The EDS line analysis indicates an insignificant composition gradient compared with
 the non-equilibrium system.

In conclusion, our comparative study of equilibrium and non-equilibrium Cu–Ni systems further
 confirms that the step flow mechanism, driven by dissolution processes such as isothermal
 solidification and solutal capillary flow, is the primary origin of slow spreading. The presence of
 pre-existing Ni in droplets or the usage of pre-saturated Cu–Ni substrates significantly influences
 the introduction of this mechanism. These findings suggest that spreading dynamics can potentially
 be controlled by tuning the occurrence of the step flow mechanism. This perspective warrants
 further exploration in a forthcoming study.

**(3) Comparison with compound-forming systems**

We agree that IMC formation at the CL can also result in a slow spreading. However, unlike the
 reaction-limited model where IMCs may completely arrest CL motion, the step flow mechanism
 proceeds through a characteristic “pinning–overflow” process that allows intermittent advancement.
 We proved this point by taking advantages of Sn/Fe systems which can change from compound-
 forming systems at 650 °C to dissolutive scenarios at 950 °C (Figure A30a). At 650 °C, we observe
 slow, linear spreading at a rate of 0.15–0.17 μm/s, with only a single instance of contact line (CL)
 overflow (Figure A30b). In-situ observations provide direct evidence that the advancing CL is
 arrested and governed by the lateral growth of a solidified edge (Figure A30(c1)-(c4)). Further SEM

analysis confirms that the edge originates from an FeSn IMC layer (Figure A30(e1)-(e4)). Spreading
 behaviors differ at 950 °C. Although the CL remains stationary during the isothermal stage, the
 droplet had already rapidly spread during the heating stage. In-situ imaging reveals the formation
 of “white meshes” ahead of the CL (Figure A30(d1)-(d4)), and SEM analysis of quenched samples
 identifies these meshes as liquid Sn spreading along surface grain boundaries (Figure A30(f1)-(f4)).
 Overall, for product forming system, the lateral IMC growth controls the spreading process. The
 linear dependence of spreading on time conforms to the reaction limited model² and the spreading
 rate is in the same order with that in reactive Sn/Au systems (less than 10 μm/s)⁷. Compared to the
 step flow mechanism, the rapid chemical reactions in Sn/Fe systems leads to a stronger solid-liquid
 interaction and consumes most Sn at the CL. Consequently, the CL can hardly overflow by
 overcoming the physical/chemical/energy barrier due to the IMC formation. For dissolutive systems,
 the spreading is fast due to a large C_R . The liquid Sn can even wet Fe surface along grain boundaries
 ahead of the droplet without an arresting CL. A more detailed discussion of grain boundary wetting
 will be presented in a forthcoming publication.

**Figure A30.** (a) Sn–Fe phase diagram showing the transition from compound-forming to dissolutive
 wetting between 650 and 950 °C. (b) CL advancement during the isothermal stage. At 650 °C (c1–
 c4), lateral growth of a ‘solidified edge’ and droplet overflow are observed, while at 950 °C (d1–
 c4),

d4) no significant edge or CL arresting. SEM images of quenched samples confirm that the
 solidified edge at 650 °C (e1–e4) is the FeSn IMC. At 650 °C (f1–f4), liquid Sn even wets Fe grain
 boundaries ahead of the droplet without CL pinning.

Another piece of evidence is from the Cu/Ti systems. The spreading is much faster for dissolutive
 systems at 1060 and 1150 °C (data for 1060 °C are from Sun et al.⁴), but significantly slower for
 product forming system at 1000 °C due to the lateral growth of Ti₂Cu layer⁴. For dissolutive systems,
 with the late presence of a step flow behavior due to a large C_R , the Cu droplets can rapidly spread
 on Ti substrates (Figure A31a). The third piece of evidence is from the comparative study between
 purely-dissolutive Sn/Bi and product forming Sn/Au systems⁵. The spreading is rapid in dissolutive
 Sn/Bi systems as a result of a large C_R . For product forming Sn/Au systems, IMC formation in the
 contact line region can arrest the triple line. Consequently, spreading is much slower and controlled
 by the lateral growth of an IMC “foot” or “film” (Figure A31b).

**Figure A31.** (a) Cu/Ti systems show rapid spreading in dissolutive regimes (1060⁴ and 1150 °C)
 but much slower spreading in the product forming regime (1000 °C⁴). (b) Comparison of dissolutive
 Sn/Bi and product forming Sn/Au systems: spreading is fast for Sn/Bi, but slower for Sn/Au due to
 lateral IMC “foot” or “film” growth⁵.

In conclusion, we have demonstrated that the step flow mechanism and the associated spreading
 behavior in the Au/Pt system are comparable to those observed in the Cu/Ni system. Accordingly,
 the Au/Pt system has been classified into the low C_L and low C_R regime in Figure A32.
 Comparative studies between equilibrium and non-equilibrium Cu/Ni systems further confirm that
 the step flow mechanism is the dominant factor for the slow spreading. Although slow spreading is
 also observed in product forming systems, we have deliberately excluded them from Figure A29, as
 their spreading behavior differs fundamentally from that governed by the step flow mechanism
 (“Linear spreading” versus. “Pinning-overflow” characteristics). We prefer to keep the primary
 focus on dissolutive systems in the revised paper. However, we acknowledge that extending the

framework to product forming systems is an interesting direction for future work.

**Related revisions 29:**

Compared with product forming systems with intermetallic compounds (IMCs) forming at the L/S
interface, dissolution is the primary interfacial activity in dissolutive systems. Dissolutive wetting
behavior can be further categorized into two main types³³: (1) purely dissolutive systems, where
dissolution is the sole liquid-substrate interaction (e.g. Sn/Bi²⁶ and Cu/Ni¹¹) (2) product forming
systems at temperatures exceeding the stability limits of all products (e.g. Cu/Ti at 1060°C⁴ and
Cu/Si at 1100°C³⁴). **(Lines 44-50 in the main text)**

With the late presence and absence of the step flow mechanism, the Cu droplets can rapidly spread
on Ti and Fe substrates due to the strong metal-metal atom affinity. This is similar to the product
forming model, where the lateral growth of IMCs at L/S interface arrests the CL and results in a
slow spreading¹⁻³. For instance, Sun et al.⁴ reported that the lateral growth of Ti₂Cu leads to a slow
spreading for the Cu/Ti system at 1000 °C. Similarly, A comparison of dissolutive Sn/Bi and product
forming Sn/Au systems shows fast spreading in Sn/Bi (conform to Case 2) but slower kinetics in
Sn/Au at 250 °C due to IMC ‘foot’ growth^{6,26}. However, spreading in Sn/Au accelerates at 450 °C,
where higher Au solubility (81 at. %) delays IMC formation⁷. Thus, early CL arrest from step flow
or IMC growth contributes to slow spreading. In contrast to the "pinning–overflow" mode of the
step flow mechanism, spreading kinetics in product forming systems is typically linear¹⁻³. **(Lines**
**307-318 in the main text)**

The proposed mechanism of dissolutive wetting is further confirmed by experimental results from
other systems. For instance, the rapid introduction of step flow mechanism in Au/Pt system aligns
well with Case 1 (Supplementary Figure A32 and Video S3). **(lines 356-359 in the main text)**

The proposed mechanism offers guidance for optimizing high-temperature processes (e.g., welding,
coating, infiltration). For example, during the high-temperature infiltration Cu–diamond composites,
diamond metallization is required to improve wettability with Cu³². Ni coatings may improve
bonding but slow infiltration, Fe coatings enable fast infiltration but weak adhesion, while Ti
coatings may balance both. **(Lines 369-373 in the main text)**

Figure A32. Dissolutive wetting behaviors map in metallic systems as a function of C_L and the C_R . (Lines 374-376 in the main text)

Reference:

1. Eustathopoulos, N. Progress in understanding and modeling reactive wetting of metals on ceramics. *Current Opinion in Solid State and Materials Science* **9**, 152–160 (2005).

2. Lin, Q. *et al.* Kinetic analysis of wetting and spreading at high temperatures: A review. *Advances in Colloid and Interface Science* **305**, 102698 (2022).

3. Dezellus, O., Hodaj, F. & Eustathopoulos, N. Chemical reaction-limited spreading: The triple line velocity versus contact angle relation. *Acta Materialia* **50**, 4741–4753 (2002).

4. Sun, Q. *et al.* Wetting of liquid copper on TC4 titanium alloy and 304 stainless steel at 1273–1433 K. *Materials & Design* **169**, 107667 (2019).

5. Yin, L., Chauhan, A. & Singler, T. J. Reactive wetting in metal/metal systems: Dissolutive versus compound-forming systems. *Materials Science and Engineering: A* **495**, 80–89 (2008).

6. Su, S., Yin, L., Sun, Y., Murray, B. T. & Singler, T. J. Modeling dissolution and spreading of Bi–Sn alloy drops on a Bi substrate. *Acta Materialia* **57**, 3110–3122 (2009).

7. Yin, L., Meschter, S. J. & Singler, T. J. Wetting in the Au–Sn System. *Acta Materialia* **52**, 2873–2888 (2004).

- 8. Brillo, J. & Egry, I. Density determination of liquid copper, nickel, and their alloys. *International*
*Journal of Thermophysics* **24**, 1155–1170 (2003).
- 9. Soda, H., McLean, A. & Miller, W. A. Surface tension measurements of liquid copper droplets
in the temperature range 1000 to 1330°C. *Transactions of the Japan Institute of Metals* **18**, 445–
454 (1977).
- 10. Eustathopoulos, N., Nicholas, M. G. & Drevet, B. *Wettability at High Temperatures*. (Pergamon,
Amsterdam ; New York, 1999).
- 11. Sun, Y. *et al.* Step flow mechanism in dissolutive wetting Cu/Ni systems. *Acta Materialia* **282**,
120519 (2025).
- 12. Cheng, Z. *et al.* High-temperature dissolutive wetting: Role of solutal marangoni convection in
multiphase transport. *Chemical Engineering Journal* **519**, 165377 (2025).
- 13. Saiz, E. & Tomsia, A. P. Atomic dynamics and marangoni films during liquid-metal spreading.
*Nature materials* **3**, 903–909 (2004).
- 14. Zheng, L., Malfliet, A., Wollants, P., Blanpain, B. & Guo, M. Effect of interfacial properties on
the characteristics and clustering of alumina inclusions in molten iron. *ISIJ International* **55**,
1891–1900 (2015).
- 15. Liu, Z., Pandelaers, L., Blanpain, B. & Guo, M. Viscosity of heterogeneous silicate melts: A
review. *Metallurgical and Materials Transactions B* **49**, 2469–2486 (2018).
- 16. Zheng, L., Malfliet, A., Wollants, P., Blanpain, B. & Guo, M. Effect of surfactant te on the
behavior of alumina inclusions at advancing solid-liquid interfaces of liquid steel. *Acta*
*Materialia* **120**, 443–452 (2016).
- 17. Wang, Z., Orejon, D., Takata, Y. & Sefiane, K. Wetting and evaporation of multicomponent
droplets. *Physics Reports* **960**, 1–37 (2022).
- 18. Lim, T. *et al.* Experimental study on spreading and evaporation of inkjet printed pico-liter droplet
on a heated substrate. *International Journal of Heat and Mass Transfer* **52**, 431–441 (2009).
- 19. Schiaffino, S. & Sonin, A. A. Molten droplet deposition and solidification at low weber numbers.
*Physics of Fluids* **9**, 3172–3187 (1997).
- 20. Alexandrov, D. V., Galenko, P. K. & Makoveeva, E. V. Solid–liquid interface stability in
solidification of a binary mixture under conductive transport and convective flow. *Journal of*

- *Applied Physics* **137**, 125110 (2025).
- 21.Kurz, W., Fisher, D. & Rappaz, M. Fundamentals of solidification. (2023).
- 22.Iveković, A. *et al.* Liquid-copper infiltration and characterization of additively manufactured W-
lattice structures. *Journal of Alloys and Compounds* **1011**, 178411 (2025).
- 23.Lin, Q., Ma, Z. & Sui, R. Wetting and pressureless infiltration behavior of Cu/316L system at
1100°C. *Surfaces and Interfaces* **42**, 103462 (2023).
- 24.Hao, X. *et al.* Microstructure and mechanical properties of dissimilar TC4 titanium alloy/304
stainless steel joint using copper filler wire. *Metall Mater Trans A* **50**, 688–703 (2019).
- 25.Liu, K., Yan, Z., Pan, R., Wang, F. & Chen, S. Effect of deposition sequence on interfacial
characteristics of inconel–copper functional bimetallic structures fabricated by directed energy
deposition-arc. *Materials Letters* **345**, 134487 (2023).
- 26.Yin, L., Murray, B. & Singler, T. Dissolutive wetting in the Bi–Sn system. *Acta Materialia* **54**,
3561–3574 (2006).
- 27.Liu, Z. *et al.* Laser 3D printing of W-cu composite. *Materials Letters* **225**, 85–88 (2018).
- 28.Lin, P.-C. *et al.* Recent advancements in copper infiltration applied to sintered steel. *Science of*
*Advanced Materials* **16**, 149–158 (2024).
- 29.Tomashchuk, I., Sallamand, P., Belyavina, N. & Pilloz, M. Evolution of microstructures and
mechanical properties during dissimilar electron beam welding of titanium alloy to stainless steel
via copper interlayer. *Materials Science and Engineering: A* **585**, 114–122 (2013).
- 30.Yost, F. G. & O’Toole, E. J. Metastable and equilibrium wetting states in the Bi–Sn system. *Acta*
*Materialia* **46**, 5143–5151 (1998).
- 31.Liashenko, O. Y. & Hodaj, F. Wetting and spreading kinetics of liquid Sn on Ag and Ag 3 Sn
substrates. *Scripta Materialia* **127**, 24–28 (2017).
- 32.Li, Y. *et al.* Transforming heat transfer with thermal metamaterials and devices. *Nat Rev Mater*
**6**, 488–507 (2021).
- 33.Singler, T. J., Su, S., Yin, L. & Murray, B. T. Modeling and experiments in dissolutive wetting:
a review. *J Mater Sci* **47**, 8261–8274 (2012).
- 34.Protsenko, P., Garandet, J.-P., Voytovych, R. & Eustathopoulos, N. Thermodynamics and kinetics
of dissolutive wetting of Si by liquid Cu. *Acta Materialia* **58**, 6565–6574 (2010).

- 35. Protsenko, P., Kozlova, O., Voytovych, R. & Eustathopoulos, N. Dissolutive wetting of Si by
molten Cu. *J Mater Sci* **43**, 5669–5671 (2008).
- 36. Saiz, E. & Tomsia, A. P. Kinetics of high-temperature spreading. *Current Opinion in Solid State
and Materials Science* **9**, 167–173 (2005).
- 37. Yin, L. *et al.* Reactive wetting in metal–metal systems. *J. Phys.: Condens. Matter* **21**, 464130
(2009).
- 38. Mortensen, A., Drevet, B. & Eustathopoulos, N. Kinetics of diffusion-limited spreading of sessile
drops in reactive wetting. *Scripta Materialia* **36**, 645–651 (1997).
- 39. Yost, F. G. Kinetics of reactive wetting. *Scripta Materialia* **42**, 801–806 (2000).
- 40. Dezellus, O. & Eustathopoulos, N. Fundamental issues of reactive wetting by liquid metals. *J
Mater Sci* **45**, 4256–4264 (2010).
- 41. Warren, R. Solid-liquid interfacial energies in binary and pseudo-binary systems. *Journal of
Materials Science* **15**, 2489–2496 (1980).
- 42. Kozlova, O., Voytovych, R., Protsenko, P. & Eustathopoulos, N. Non-reactive versus dissolutive
wetting of ag–cu alloys on cu substrates. *Journal of materials science* **45**, 2099–2105 (2010).
- 43. Xian, A.-P. Precursor film of tin-based active solder wetting on ceramics. *Journal of Materials
Science* **28**, 1019–1030 (1993).
- 44. Blake, T. D. & Haynes, J. M. Kinetics of liquid/liquid displacement. *J. Colloid Interface Sci.* **30**,
421 (1969).
- 45. Hoffman, R. L. A study of the advancing interface: II. Theoretical prediction of the dynamic
contact angle in liquid-gas systems. *Journal of Colloid and Interface Science* **94**, 470–486 (1983).
- 46. Hansen, R. J. & Toong, T. Y. Dynamic contact angle and its relationship to forces of
hydrodynamic origin. *Journal of Colloid and Interface Science* **37**, 196–207 (1971).
- 47. Zhao, J.-C. *Methods for Phase Diagram Determination*. (elsevier, 2011).